# Foundation Models Meet Federated Learning: A One-shot Feature-sharing Method with Privacy and Performance Guarantees

**Mahdi Beitollahi**                                      *mahdi.beitollahi@huawei.com*
*Huawei Noah's Ark Lab, Montreal, Canada.*

**Alex Bie**                                              *alex.bie@huawei.com*
*Huawei Noah's Ark Lab, Montreal, Canada.*

**Sobhan Hemati**                                         *sobhan.hemati@huawei.com*
*Huawei Noah's Ark Lab, Montreal, Canada.*

**Leo Maxime Brunswic**                                   *sobhan.hemati@huawei.com*
*Huawei Noah's Ark Lab, Montreal, Canada.*

**Xu Li**                                                 *xu.lica@huawei.com*
*Huawei Technologies Canada Inc., Ottawa, Canada.*

**Xi Chen**                                               *xi.chen4@huawei.com*
*Huawei Noah's Ark Lab, Montreal, Canada.*

**Guojun Zhang**                                          *guojun.zhang@huawei.com*
*Huawei Noah's Ark Lab, Montreal, Canada.*

**Reviewed on OpenReview:** *https://openreview.net/forum?id=55593xywWG*

## Abstract

Adapting foundation models for downstream tasks via Federated Learning (FL) is a promising strategy for protecting privacy while leveraging the capability of foundation models. However, FL's iterative training and model transmission result in high communication costs and GPU memory demands, making large foundation models impractical for FL. This paper introduces a one-shot FL method with a server-side performance bound to enable foundation models by reducing communication costs and GPU memory requirements. Our approach, FedPFT (FL with *P*arametric *F*eature *T*ransfer), involves clients learning and transferring parametric models for features extracted from frozen foundation models in a single round. Parametric models are then used to generate synthetic features at the server to train a classifier head. We evaluate FedPFT across eight vision datasets using three vision foundation models. Our findings demonstrate that FedPFT is agnostic to data heterogeneity and network topology and it enhances the communication-accuracy frontier up to 7.8%. Finally, we show FedPFT's compatibility with differential privacy and its resilience against reconstruction attacks. Our work highlights the capability of private, feature-sharing methods for one-shot knowledge transfer using foundation models.

## 1 Introduction

Federated learning (FL) (McMahan et al., 2017) is a learning paradigm that can facilitate fine-tuning foundation models on downstream tasks across clients without sharing their raw data to adhere to privacy concerns and regulatory guidelines such as GDPR (European Parliament & Council of the European Union). Traditional FL takes place iteratively over multiple rounds, wherein each client *trains* a local model with its

data and subsequently *transmits* it to a central server for aggregation. The frequent exchange or training of models, each containing hundreds of millions of parameters, imposes an intolerable burden of high communication costs and GPU memory requirement on each client, rendering the scalability of cross-device FL systems to large foundation models impractical (Konečný et al., 2016; Beitollahi & Lu, 2023).

*One-shot FL* (Guha et al., 2019) is an approach to address the high communication cost of FL by requiring a single communication round. Moreover, one-shot FL addresses several drawbacks of multi-round FL: a) coordinating a multi-iteration FL process across a large number of clients is susceptible to failures, stemming from issues such as client dropout, resource heterogeneity, and real-world implementation challenges; b) support of scenarios where multi-round FL is impractical, e.g., dynamic environments (Zhou, 2022) where the global model is required to adapt to evolving environments; c) frequent communication poses a higher chance of being intercepted by outsider attacks such as man-in-the-middle attacks (Bouacida & Mohapatra, 2021).

While one-shot FL holds promise, current methods lack the accuracy of multi-round FL counterparts (see Figure 1). Furthermore, convergence analysis of multi-round FL cannot be extended to one-shot FL due to its reliance on a large number of iterations; thus, current one-shot FL methods lack theoretical guarantees. Additionally, one-shot FL alone cannot mitigate the GPU memory demands.

Recent strides in foundation models (Bommasani et al., 2021) present novel opportunities for training-free and communication-efficient knowledge sharing among clients in FL. Foundation models such as CLIP (Radford et al., 2021) and GPT series (Radford et al., 2019; Brown et al., 2020) demonstrate remarkable performance via task-agnostic representations. Features from these models can be leveraged "as they are" without extensive fine-tuning, via probing or clustering, to achieve performance on downstream tasks surpassing that of complex, task-specific models. Further, in well-trained foundation models, there exist linear paths in representation space that vary according to semantic axes (Mikolov et al., 2013; Härkönen et al., 2020). A corollary is that fitting simple, smooth, parametric distributions in representation space can lead to realistic samples sharing semantic characteristics; on the other hand, this is certainly not true in input space (i.e., Gaussians cannot model the distribution of natural images).

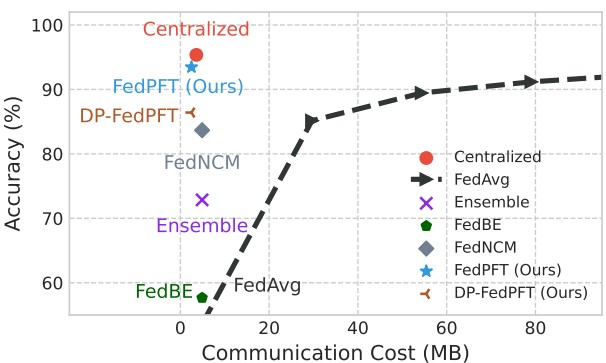

**Figure 1:** Comparison of different one-shot FL methods for image classification on Caltech-101 with 50 clients. See Section 5.2 for experimental details. FedPFT and DP-FedPFT outperform other one-shot FL methods and are competitive with transmitting real features (Centralized). With more communication budget, multi-round FL (i.e. FedAvg) performs better than one-shot methods.

**Our contributions.** In this paper, we propose **FedPFT** (**Fed**erated Learning with **P**arametric **F**eature **T**ransfer) a training-free, one-shot FL method with a server-side guarantee to enable foundation models in FL. In FedPFT, each client learns and transfers parametric models from extracted features from pre-trained, frozen foundation models in one round. These parametric models are used to generate synthetic features at the server to train a classifier head (as illustrated in Figure 2). Therefore, FedPFT does not require training or fine-tuning of large foundation models. We evaluate FedPFT with different families of parametric models, including a mixture of Gaussians. Our evaluation of eight vision datasets with three vision foundation models shows that FedPFT achieves performance close to centralized training and is agnostic to data heterogeneity. We show that FedPFT supports both centralized and decentralized FL. Finally, we provide theoretical server-side performance guarantees for FedPFT.

Our main contributions include:

- We introduce FedPFT, a training-free, one-shot parametric feature-sharing framework that enables foundation models in FL to enhance the communication-accuracy frontier.

- We evaluate FedPFT using eight vision datasets and three vision foundation models, showing that FedPFT enhances the communication-accuracy frontier by up to 7.8% and is agnostic to various network topologies and distribution shifts including label shift and task shift.

- We extend FedPFT to offer differential privacy guarantees, demonstrating favorable privacy-accuracy tradeoffs. Additionally, we conduct reconstruction attacks on various feature-sharing schemes and demonstrate the privacy risks of sending real features.

- We theoretically analyze the performance of FedPFT and provide a server-side bound.

## 2 Related work

**One-shot FL.** Naive parameter averaging methods such as FedAvg (McMahan et al., 2017) do not perform well in one-shot FL settings (Guha et al., 2019) (see Figure 1). To overcome this limitation, researchers have explored diverse strategies, including knowledge distillation (KD) (Hinton et al., 2015), ensemble learning (Guha et al., 2019; Chen & Chao, 2020), and generative method (Kasturi & Hota, 2023; Zhang et al., 2022). KD methods commonly rely on a *public dataset* for distillation which is not always available in practice. Further, ensemble learning methods use client models collectively as the global model, but as we demonstrate, they perform poorly with extremely diverse data due to *overfitting*. Finally, generative methods are used to create or extract synthetic images at the server. Generative methods require extensive *training* on the client and can be challenging when clients access only a few samples. In our experiments, we were unsuccessful in utilizing methods like DENSE (Zhang et al., 2022) to extract synthetic images from frozen, pre-trained models (see Appendix 5 for details).

**Foundation models for FL.** Foundation models can greatly enhance the training of FL. For instance, clients can directly conduct fine-tuning on their local data without training from scratch (Chen et al., 2022; Nguyen et al.; Tan et al., 2022) to converge faster and achieve better performance. Further, foundation models allow a novel sharing paradigm. For instance, clients can train and share prompts (Zhao et al., 2023; Guo et al., 2023), instead of sharing high-dimensional model parameters. However, these methods require multiple rounds of sharing parameters and training of the foundation models.

**Sharing feature statistics in FL.** Sharing feature statistics has been explored in FL. For instance, CCVR (Luo et al., 2021) and FedImpro (Tang et al., 2024), in addition to conducting a multi-iteration FL, represent the features of each class as a Gaussian distribution and transmit both the model parameters and the distribution parameters to the server to sample and re-train the classifier. Similarly, FedNCM (Legate et al., 2024) and FedPCL (Tan et al., 2022) consider sharing class prototypes for learning the classifier head using Nearest Class Means and Contrastive Learning, respectively. However, neither a Gaussian nor a class prototype can completely represent the features (see Section 6.1), FedPFT outperforms FedNCM and CCVR by up to 7% (see Figure 4). Further, none of these methods provide any theoretical performance or privacy analysis for the effect of sharing feature statistics. Our work is an extension of these methods to formally study and analyze feature-sharing in training-free, one-shot FL settings. Table 1 highlights the differences between FedPFT and similar approaches that rely on sending the statistics of features.

**Table 1:** Comparison between FedPFT and approaches that rely on sending the statistics of features.

| Methods | Parameterization of features | Performance guarantee | One-shot | Training-free foundation model | Privacy guarantee | Supports decentralized FL |
|---|---|---|---|---|---|---|
| CCVR (Luo et al., 2021) | Gaussian | ✗ | ✗ | ✗ | ✗ | ✗ |
| FedImpro (Tang et al., 2024) | Gaussian | ✗ | ✗ | ✗ | ✗ | ✗ |
| FedPCL (Tan et al., 2022) | Class prototypes | ✗ | ✗ | ✓ | ✗ | ✗ |
| FedNCM (Legate et al., 2024) | Class prototypes | ✗ | ✓ | ✓ | ✗ | ✗ |
| FedPFT (Ours) | Mixture of Gaussians | ✓ | ✓ | ✓ | ✓ | ✓ |

## 3 Preliminaries

**Federated learning.** The objective of one-shot FL is to learn a model $w$ from data distributed across $I$ clients, each communicating only once. We represent $D_i$ as the local dataset for client $i \in \{1, ..., I\}$ of example-label pairs $(\mathbf{x}, y)$, and we denote $n_i := |D_i|$ as the number of samples for local data of client $i$.

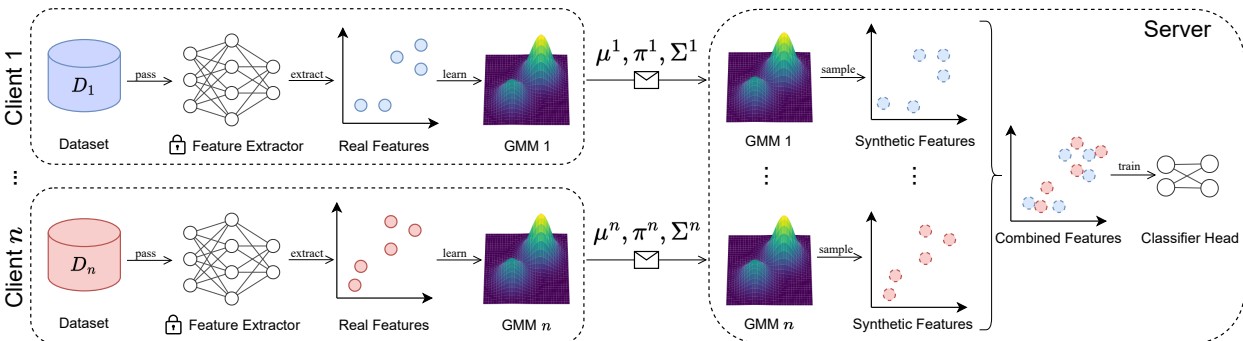

**Figure 2:** Illustration of FedPFT in centralized FL. Each client learns GMMs for their distributions of extracted features for each class. Then, GMM's parameters are transmitted to the server, which then samples from these distributions to train a classifier head as the global model.

The model $w$ is decomposed into $w := h \circ f$, where $f$ represents a *feature extractor* mapping input $\mathbf{x}$ to a $d$-dimensional embedding, and $h : \mathbb{R}^d \to \mathbb{R}^C$ is the classifier head (i.e., linear layer), with $C$ denoting the number of classes. In FL setups, the goal is to minimize the objective function:

$$L(w) := \sum_{i=1}^{I} \frac{n_i}{n} \mathbb{E}_{(\mathbf{x},y)\sim D_i}[\ell(w; \mathbf{x}, y)], \tag{1}$$

where $\ell$ is the cross-entropy loss function and $n := \sum_{i=1}^{I} n_i$.

**Gaussian mixture models.** We employ Gaussian mixture models (GMMs) as one of our chosen parametric models, given their concise parameterization and status as universal approximators of densities (Scott, 2015). Our approach relies on learning Gaussian mixtures over feature space $\mathbb{R}^d$. Let $\mathbb{S}^+$ denote the set of all $d \times d$ positive definite matrices. We denote by $\mathcal{G}(K)$ the family of all Gaussian mixture distributions comprised of $K$ components over $\mathbb{R}^d$. Each density function $g \in \mathcal{G}(K)$ is identified by a set of tuples $\{(\pi_k, \boldsymbol{\mu}_k, \boldsymbol{\Sigma}_k)\}_{k=1}^{K}$, where each mixing weight $\pi_k \geq 0$ with $\sum_{k=1}^{K} \pi_k = 1$, each mean vector $\boldsymbol{\mu}_k \in \mathbb{R}^d$, and each covariance matrix $\boldsymbol{\Sigma}_k \in \mathbb{S}^+$, satisfying:

$$g := \sum_{k=1}^{K} \pi_k \cdot \mathcal{N}(\boldsymbol{\mu}_k, \boldsymbol{\Sigma}_k) \tag{2}$$

where $\mathcal{N}(\boldsymbol{\mu}, \boldsymbol{\Sigma})$ refers to the Gaussian density over $\mathbb{R}^d$ with mean $\boldsymbol{\mu}$ and covariance $\boldsymbol{\Sigma}$. In addition, we denote $\mathcal{G}_{\text{diag}}(K)$ to denote Gaussian mixtures comprising of diagonal Gaussians, i.e., with the additional constraint that all $\boldsymbol{\Sigma}_k$ are diagonal. We also denote $\mathcal{G}_{\text{spher}}$ to Gaussians mixtures with spherical covariances, i.e., each $\boldsymbol{\Sigma}_{\boldsymbol{k}} \in \{\lambda \mathbb{I}_d : \lambda \in \mathbb{R}_{\geq 0}\}$. We may also refer to the family of full covariance $\mathcal{G}(K)$ as $\mathcal{G}_{\text{full}}(K)$, and use $\mathcal{G}_{\text{cov}}(K)$ to denote different family types.

## 4    Methods

In this section, We first describe FedPFT for the centralized FL setting, assuming the presence of a centralized server connected to all clients. We also describe the modifications required to adapt FedPFT for (a) the decentralized FL setting; and (b) differential privacy requirements. Finally, We provide a performance guarantee for FedPFT.

### 4.1    Centralized FedPFT

In this conventional FL setup, a central server can aggregate the knowledge from all clients. In the centralized FedPFT scenario, as illustrated in Figure 2, each client $i$ extracts class-conditional features from its local dataset for each available class $c \in \{1, ..., C\}$:

$$F^{i,c} := \{f(\mathbf{x}); (\mathbf{x}, y) \in D_i, y = c\}, \tag{3}$$

using the pre-trained foundation model feature extractor $f$. Next, each client $i$ learns runs the Expectation Maximization (EM) algorithm (Dempster et al., 1977) on $F^{i,c}$ to learn a GMM $g^{i,c} \in \mathcal{G}_{\mathrm{cov}}(K)$ for each class $c$ that approximates $F^{i,c}$. Finally, each client sends its $g^{i,c}$ parameters $\{(\pi_k^{i,c}, \boldsymbol{\mu}_k^{i,c}, \boldsymbol{\Sigma}_k^{i,c})\}_{k=1}^K$ to the server. On the server side, the server samples class-conditional synthetic features $\tilde{F}^{i,c}$ from each received $g^{i,c}$ parameters, i.e,

$$\tilde{F}^{i,c} \sim g^{i,c} = \sum_{k=1}^K \pi_k^{i,c} \cdot \mathcal{N}(\boldsymbol{\mu}_k^{i,c}, \boldsymbol{\Sigma}_k^{i,c}) \tag{4}$$

with the size of $|F^{i,c}|$. Then, the server combines class-conditional synthetic features $\tilde{F}^{i,c}$ from all the clients and classes to create synthetic feature dataset $\tilde{F}$ as follows,

$$\tilde{D} = \bigcup_{i=1}^I \bigcup_{c=1}^C \{(\mathbf{v}, c) : \mathbf{v} \in \tilde{F}^{i,c}\}. \tag{5}$$

Finally, the server trains a classifier head $h$ on $\tilde{F}$, minimizing $\mathbb{E}_{(\mathbf{v},y)\sim\tilde{D}}[\ell(h; \mathbf{v}, y)]$ where $\ell$ is the cross-entropy loss. The trained, global classifier head $h$ is then sent back to the clients, and clients can use $w = h \circ f$ as the global model. This process is described in Algorithm 1.

## 4.2 Decentralized FedPFT

Traditional FL relies on a central server for coordination, which introduces a single point of failure and limits scalability. This centralized architecture also poses privacy and security risks by exposing model updates and reduces fault tolerance by making the system dependent on server availability. Decentralized FL mitigates these issues by enabling direct peer-to-peer communication, improving resilience, and eliminating the need for a trusted aggregator. Therefore, we extend FedPFT to the decentralized setting.

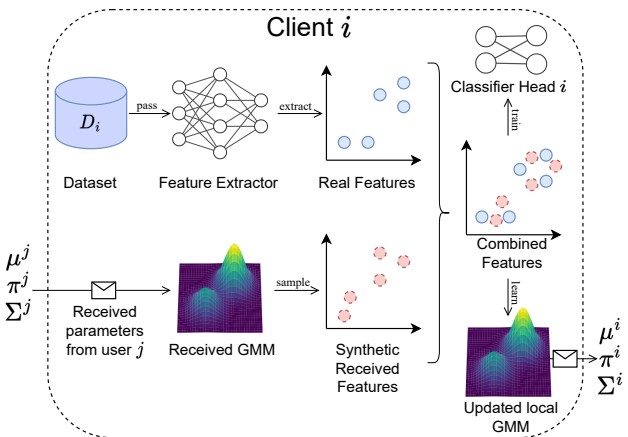

**Figure 3:** Illustration of decentralized FedPFT where clients update received GMMs with their local data.

In decentralized FL, there is no centralized server, and clients are all connected in an ad-hoc manner. Therefore, each client has the responsibility of aggregating the knowledge of its dataset with other clients and passing the knowledge to the next client. FedPFT for decentralized FL is illustrated in Figure 3. In this method, similar to centralized FL, each client $i$ creates class conditional features from its local dataset for each label $c$, i.e., $F^{i,c}$. Then, client $i$ samples from the received GMMs from client $j$ to generate synthetic class-conditional features $\tilde{F}^{j,c}$. Next, client $i$ runs the Expectation Maximization (EM) algorithm on $F^{i,c} \cup \tilde{F}^{j,c}$ to learn a GMM $g^{i,c} \in \mathcal{G}_{\mathrm{cov}}(K)$ for each class $c$ that approximates the union of $F^{i,c}$ and $\tilde{F}^{j,c}$. The parameters of $g^{i,c}$ are sent to the next client. At the same time, each client can use the combined features $F^{i,c} \cup \tilde{F}^{j,c}$ to train its local classifier head $h_i$. By passing GMMs between clients, knowledge of each client is accumulated and propagated between clients with just one round of communication where the last client has the knowledge of all the clients.

## 4.3 Differential Privacy

FedPFT's goal is to transfer the parameters of GMMs without leaking clients' private information in their dataset. For formal privacy guarantees, we employ differential privacy (DP, Dwork et al., 2006). To make FedPFT differentially private, we use the Gaussian mechanism (Dwork et al., 2006; 2014) to privatize the release of all mean vectors and covariance matrices. The following theorem along with the proof in the Appendix E provides the $(\epsilon, \delta)$-differential privacy guarantee for FedPFT in the case of Gaussians $(\mathcal{G}_{\mathrm{full}}(K = 1))$.

**Theorem 4.1** (**Privacy Guarantee**). *Suppose the feature embedding $f$ satisfies $\|f\|_2 \leq 1$. Let $\hat{\boldsymbol{\mu}}(\cdot)$ and $\hat{\boldsymbol{\Sigma}}(\cdot)$ be the estimator of mean and covariance, respectively. Define the Gaussian mechanism*

$$\mathcal{M} : D \mapsto (\widetilde{\boldsymbol{\mu}}(D), \widetilde{\boldsymbol{\Sigma}}(D)), \tag{6}$$

$$\widetilde{\boldsymbol{\mu}}(D) = \hat{\boldsymbol{\mu}}(f(D)) + \Delta\boldsymbol{\mu}, \tag{7}$$

$$\widetilde{\boldsymbol{\Sigma}}(D) = \mathrm{Proj}_{\mathbb{S}_+}(\hat{\boldsymbol{\Sigma}}(f(D)) + \Delta\boldsymbol{\Sigma}), \tag{8}$$

*where the elements of vector $\Delta\boldsymbol{\mu}$ and matrix $\Delta\boldsymbol{\Sigma}$ are sampled from independent $\mathcal{N}\left(0, \left(\frac{4}{n_i \epsilon}\sqrt{5\ln(4/\delta)}\right)^2\right)$, and $\mathrm{Proj}_{\mathbb{S}_+}$ is the projection onto the set of positive semi-definite matrices. Then, the Gaussian mechanism $\mathcal{M}$ satisfies $(\epsilon, \delta)$-differential privacy.*

Note that the assumption of normalized features in Theorem 4.1 does not limit the performance of networks with soft-max loss function since both normalized and unnormalized features have the same expressive power as shown in Proposition 3.A in (Zhang et al., 2023). Further, note that Theorem 4.1 only assures the privacy of GMMs. However, FedPFT sends label counts along with the GMMs to the server, which may not be advisable from a privacy perspective. In Section A.3, we discuss cryptographic adaptations that can be made to address this issue.

### 4.4 Performance Guarantee

FedPFT trains a classifier head on the server using synthetic features. The training loss at the server assures that the classifier can classify synthetic features effectively. But how can we guarantee that this classifier works well for the real features of clients without accessing real features? The following theorem provides a server-side bound on the performance of the classifier head on the real features:

**Theorem 4.2** (**Performance Guarantee**). *For any classifier head $h$, with $\ell^{0-1}$ and $\widetilde{\ell^{0-1}}$ the 0-1 losses of $h$ on $\bigcup_{i=1}^{I} f(D_i)$ and $\bigcup_{i=1}^{I}\bigcup_{c=1}^{C} \widetilde{F}^{i,c}$ respectively,*

$$\underbrace{\ell^{0-1}}_{\textit{Raw feature's 0-1 loss}} \leq 2 \underbrace{\widetilde{\ell}^{0-1}}_{\substack{\textit{Synthetic feature's} \\ \textit{0-1 loss}}} + \sum_{i=1}^{I} \mathbb{E}_c \underbrace{\left[L_{i,c}^{EM}\right]}_{\substack{\textit{EM loss of fitting} \\ \textit{a GMM on class c} \\ \textit{of client i}}}, \tag{9}$$

*where $L_{i,c}^{EM} = \frac{1}{\sqrt{2}}\sqrt{\mathcal{H}_{i,c} - \ell_{i,c}^{EM}}$, and $\ell_{i,c}^{EM}$ is the class-wise log-likelihood of the EM, and $\mathcal{H}_{i,c}$ is the self-entropy of the distribution $F^{i,c}$.*

This theorem states that the 0-1 loss on all the real features (centralized loss) is bounded by twice the 0-1 loss of the server on all the generated synthetic features plus the sum of the expectation of EM loss of fitting GMMs on all the classes across all the clients. Therefore, if the goal is to have a centralized 0-1 loss of 0.1, the 0-1 loss on synthetic features should be at most 0.05 given perfect GMMs (i.e. $L^{EM} = 0$). Notably, this bound is a *server-side guarantee* since $\widetilde{\ell}^{0-1}$ can be calculated at the server and $\mathbb{E}_c[L_{i,c}^{EM}]$ is already calculated during learning of GMMs at the clients and can be sent to the server. Using this theorem, the server can calculate the performance of clients without accessing their data. Complete proof of this theorem along with a comparison to other bounds can be found in Appendix F.

## 5 Experiments

We examine the knowledge transfer capabilities of FedPFT and DP-FedPFT in the one-shot setting, comparing them with state-of-the-art methods across various data heterogeneity settings and network topologies. Our experiments support the claim that FedPFT: (1) compares favorably against existing one-shot FL methods; (2) succeeds in a variety of extreme client distribution scenarios challenging for FL; and (3) supports decentralized network topologies.

**Table 2:** Summary of datasets and foundation models

| Dataset | Min size | $|D|$ | $C$ | Foundation Model | $d$ |
|---|---|---|---|---|---|
| CIFAR10 | 32 | 50k | 10 | ResNet-50 | 2048 |
| CIFAR100 | 32 | 50k | 100 | ResNet-50 | 2048 |
| PACS (P) | 224 | 1.3k | 7 | ViT-B/16 | 768 |
| PACS (S) | 224 | 3k | 7 | ViT-B/16 | 768 |
| Office Home (C) | 18 | 3.4k | 65 | ViT-B/16 | 768 |
| Office Home (P) | 63 | 3.5k | 65 | ViT-B/16 | 768 |
| Caltech101 | 200 | 6k | 101 | CLIP, ViT-B/32 | 768 |
| Stanford Cars | 240 | 12k | 196 | CLIP, ViT-B/32 | 768 |
| Oxford Pets | 108 | 3.6k | 37 | CLIP, ViT-B/32 | 768 |
| Food101 | 512 | 75k | 101 | CLIP, ViT-B/32 | 768 |

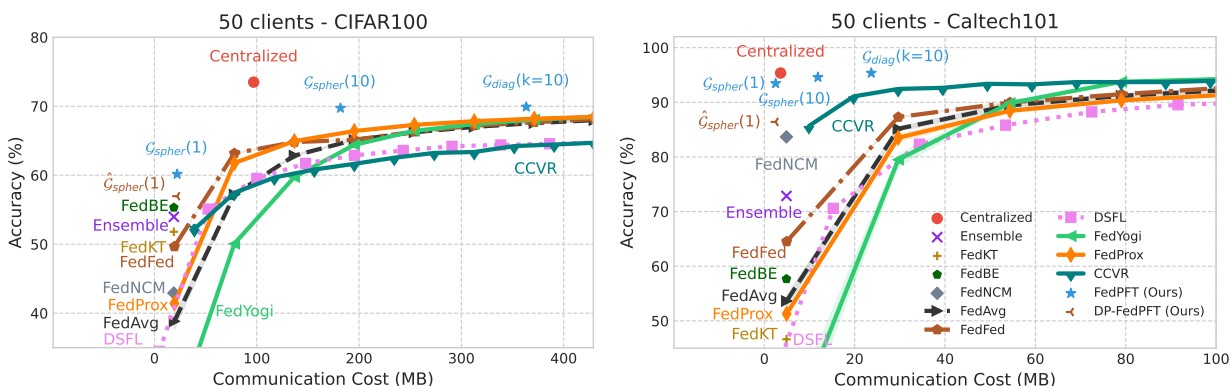

**Figure 4:** FedPFT vs existing one-shot and multi-round FL methods in Centralized setting with CIFAR100 (left) and Caltech 101 (right) dataset. FedPFT ($\mathcal{G}$) and DP-FedPFT ($\hat{\mathcal{G}}$) surpass other one-shot FL methods, and are competitive with sending raw features (Centralized).

## 5.1 Experimental setting

**Datasets and foundation models.** We use 8 vision datasets including CIFAR10/100 (Krizhevsky et al., 2014), PACS (Li et al., 2017), Office Home (Venkateswara et al., 2017), Caltech101 (Li et al., 2022), Stanford Cars (Krause et al., 2013), Oxford Pets (Parkhi et al., 2012), and Food101 (Bossard et al., 2014) and three foundation models including ResNet-50 (He et al., 2016) pre-trained on ImageNet, ViT-B/16 (Dosovitskiy et al., 2020) pre-trained on ImageNet, and the CLIP (Radford et al., 2021) image encoder as our feature extractor $f$. Table 2 provides a summary of the datasets and the corresponding feature extractors used. We keep the foundation models frozen for all the experiments except in Table 6 where we analyze the effect of fine-tuning.

**Implementation and Baselines.** We use FedPFT to train the classifier head of foundation models. We provide a discussion on training other layers in Appendix A.2. Also, We provide the code for implementing GMMs in FedPFT in Appendix D. For transmitting GMM parameters, we use a 16-bit encoding. We use $\mathcal{G}_{\mathrm{cov}}(K)$ and $\hat{\mathcal{G}}_{\mathrm{cov}}(K)$ to denote FedPFT and DP-FedPFT, respectively, where *cov* represented covariance types. We set $\epsilon = 1$, $\delta = 1/|D^{i,c}|$, and use $K = 1$ mixture components for all DP-FedPFT experiments since our theoretical analysis currently supports only this case. We run all the experiments for three seeds and report the mean and standard deviation of the accuracy on the hold-out test dataset, along with its communication cost. Further experimental details can be found in Appendix G. For baselines, we present the results of centralized training with raw, pre-trained features (*Centralized*) as the oracle. We also include the results of an *Ensemble* comprising locally trained classifier heads from clients, where the class with the highest probability across the models is selected.

**Table 3:** FedPFT in three extreme shifts in two-client decentralized FL. We format **first** and oracle results.

| Method | Disjoint Label shift | | Covariate shift | | Task shift | |
|---|---|---|---|---|---|---|
| | CIFAR-10 | CIFAR-100 | PACS (P→S) | Office (C→P) | Caltech101 ↓ Stanford Cars | Oxford Pets ↓ Food101 |
| Centralized | 90.85 ± 0.03 | 73.97 ± 0.06 | 89.15 ± 0.17 | 82.00 ± 0.16 | 81.88 ± 0.06 | 88.48 ± 0.05 |
| Ensemble | 80.18 ± 0.30 | 57.94 ± 0.22 | 79.59 ± 0.83 | 71.36 ± 0.46 | 58.33 ± 2.01 | 83.54 ± 0.21 |
| Average | 77.66 ± 1.04 | 56.82 ± 0.22 | 77.83 ± 0.23 | 69.69 ± 0.69 | 72.65 ± 0.13 | 83.25 ± 0.59 |
| KD | 74.22 ± 0.42 | 55.62 ± 0.20 | 67.69 ± 2.47 | 72.15 ± 0.75 | 40.46 ± 0.31 | 43.67 ± 0.04 |
| $\mathcal{G}_{\text{diag}}$(K=10) | 86.19 ± 0.15 | 69.97 ± 0.07 | **89.12** ± 0.18 | 80.56 ± 0.31 | 81.74 ± 0.06 | 88.22 ± 0.07 |
| $\mathcal{G}_{\text{diag}}$(K=20) | **86.89** ± 0.02 | **70.31** ± 0.10 | 89.00 ± 0.15 | **80.94** ± 0.16 | **81.75** ± 0.04 | **88.25** ± 0.08 |

## 5.2 Comparing to existing one-shot and multi-round FL methods

In Figure 4, we compare the performance of FedPFT with state-of-the-art one-shot FL methods in centralized settings with 50 clients for training the classifier head. We conduct tests on both CIFAR100 with the ResNet-50 feature extractor and the Caltech101 dataset with the CLIP ViT feature extractor, where the samples are distributed across clients according to Dirichlet ($\beta = 0.1$). The benchmarked multi-round methods include FedAvg (McMahan et al., 2017), FedProx (Li et al., 2020b), DSFL (Beitollahi et al., 2022), FedFed (Yang et al., 2023), CCVR (Luo et al., 2021) and FedYogi (Reddi et al., 2020), along with single-shot methods such as FedBE (Chen & Chao, 2020), FedNCM (Legate et al., 2024), FedKT (Li et al., 2020a), and Ensemble. We were unsuccessful in running model inversion methods including DENSE (Zhang et al., 2022) for frozen, pre-trained models (See Appendix A.4 for further details).

Figure 4 shows that FedPFT and DP-FedPFT beat other one-shot FL methods; and are competitive with sending raw features (Centralized). With more communication budget, multi-round FL methods perform better than existing one-shot methods. Additionally, Figure 4 demonstrates different tradeoffs for varying numbers of mixtures $K$ and covariance types. See Section 6.1 for more discussion on these tradeoffs.

## 5.3 Extreme shifts in peer-to-peer decentralized FL

By design, FedPFT is agnostic to data heterogeneity. To assess this claim, we examine decentralized FL settings with two clients: source and destination. These clients exhibit significantly different training distributions. Specifically, we explore three types of extreme shift scenarios—label shift, covariate shift, and task shift—between clients, where the source can communicate only once to the destination client. We report the performance of the destination's trained classifier head on both clients' test datasets in Table 3. Table 3 demonstrates that FedPFT succeeds in various extreme client distribution scenarios and highlights the limitations of vanilla averaging, ensembling, and KD methods. Below we provide the details of the benchmarks and distribution shifts. Further details of this experiment including the communication cost can be found in Appendix G.4.

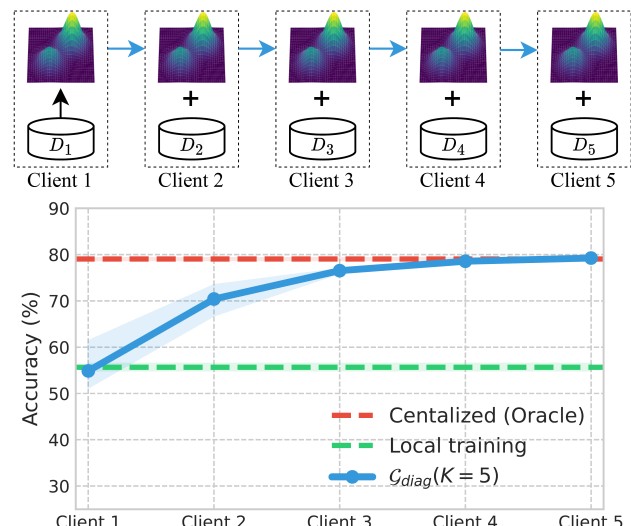

**Figure 5:** (top) Five clients in a linear topology. Each client updates its received GMM with its local data and sends it to the next client. (bottom) Results of FedPFT with 5 clients in linear topology as illustrated in Figure

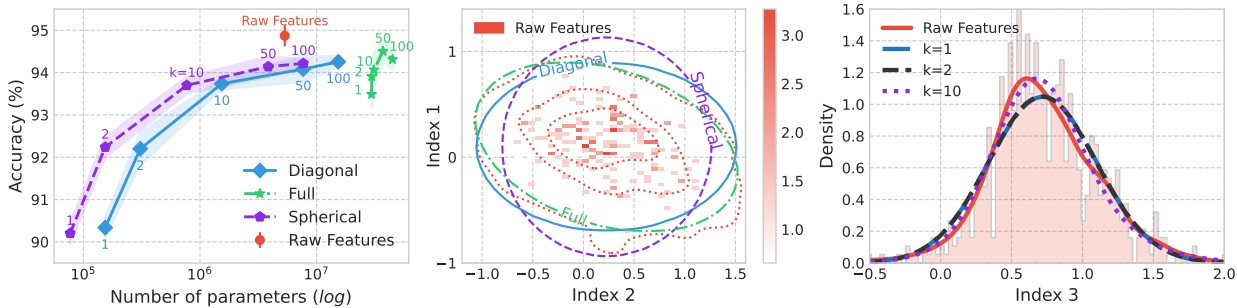

**Figure 6:** Comparing real and synthetic feature distributions using the Caltech101 dataset. (Left) Classifier accuracy on raw vs. synthetic features from various GMMs. (Middle) 2-dimensional and (Right) 1-dimensional distribution of random indexes of real features vs GMMs' counterparts with different covariance types and number of mixtures.

### 5.4 Linear topology

In this setup, we demonstrate the propagation and accumulation of knowledge using FedPFT in decentralized FL with five clients in a linear topology where each client has access to 100 i.i.d. samples of the CIFAR10 dataset, as illustrated in Figure 5 (top). Using GMMs, we transfer knowledge of client 1 to client 5 in four communication rounds and report the performance of each client's classifier head trained on its received GMM on the entire 5 client dataset in Figure 5 (bottom). We also compare FedPFT with centralized training and local training, where we train a classifier head for each client's dataset. Figure 5 (bottom) shows that as GMMs pass through clients, they can accumulate and propagate the knowledge and achieve performance close to 1.8% of centralized training.

## 6 Analysis

In this section, we provide a comprehensive analysis of FedPFT, examining its accuracy, communication cost, and privacy characteristics. In particular, we: (1) examine the tradeoff between accuracy and communication cost for different families of GMMs; (2) prove theoretical guarantees on local client accuracy; (3) estimate communication costs; and (4) analyze privacy leakage in FedPFT, compared to sending raw features.

### 6.1 How well do GMMs model feature distributions?

FedPFT relies on learning the distribution of features of each class using GMMs. We aim to assess how effectively GMMs can estimate class-conditional features. We measure the accuracy gap between two classifier heads: one trained on real features and another on synthetic features generated using GMMs. Monitoring the accuracy gap helps us evaluate the discriminative power of synthetic features. We examine the effect of the number of mixtures $K$ and the type of covariance matrix. For this experiment, pictured in Figure 6, we use CLIP ViT-B/32 features on the Caltech101 dataset. For various families of GMMs, we plot accuracy and the total number of statistical parameters.

Figure 6 (left) shows that 10-50 Gaussians are sufficient to represent raw extracted features with less than a 1% drop in accuracy. Notably, GMMs with a spherical covariance matrix exhibit better tradeoffs between communication and accuracy compared to full/diagonal covariance matrices. Additionally, we illustrate the 1-dimensional (Figure 6 right) and 2-dimensional (Figure 6 middle) density of random indexes of raw features and compare them with GMMs.

### 6.2 Communication cost of FedPFT

Here, we estimate the communication cost of FedPFT and compare it to sending the classifier head or sending the raw features. Denoting by $d$ the feature dimension, $K$ the number of components, and $C$ the number of

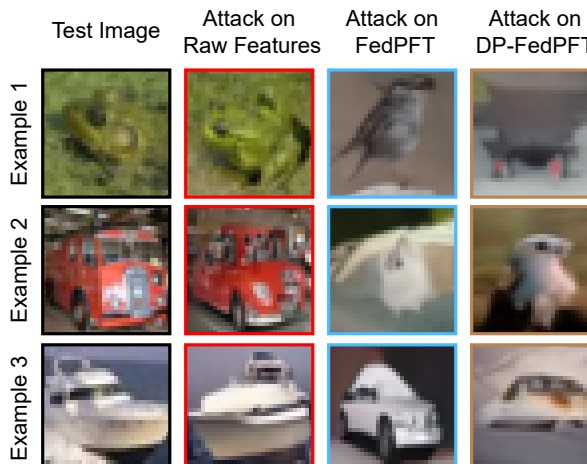

**Figure 7:** (left) Results of reconstruction attacks on feature-sharing schemes using three random test images from CIFAR-10. Attackers can reconstruct raw features (middle-left) to generate images resembling real data (left). However, the same reconstruction on FedPFT (middle-right) and DP-FedPFT (right) does not resemble the real image. *For more reconstruction examples see Figures 12 and 13 in the Appendix.* (right) Quantitative metrics for *set-level reconstruction.* We report image similarity on the top 1% ($n = 90$) of test set images, by their SSIM to a member of the reconstruction set. As a baseline, we report the results of treating the train set as a reconstruction set. For further elaboration on our threat model, see Appendix H.

| Reconstruction source | Similarity measure | | |
|---|---|---|---|
| | $PSNR \uparrow$ | $LPIPS \downarrow$ | $SSIM \uparrow$ |
| Raw features | 16.5 | .257 | .535 |
| FedPFT | 14.7 | .423 | .508 |
| DP-FedPFT | 14.6 | .511 | .483 |
| Train set | 14.7 | .389 | .500 |

classes, we calculate the communication cost of FedPFT for different covariance families:

$$\text{Cost}(\mathcal{G}_{\text{full}}) : (2d + \frac{d^2 - d}{2}) + 1)KC \sim \mathcal{O}(d^2CK), \tag{10}$$

$$\text{Cost}(\mathcal{G}_{\text{diag}}) : (2d + 1)KC \sim \mathcal{O}(2dCK), \tag{11}$$

$$\text{Cost}(\mathcal{G}_{\text{spher}}) : (d + 2)KC \sim \mathcal{O}(dCK). \tag{12}$$

Equations (10 - 12) indicate that the communication cost of FedPFT is independent of the number of samples of each client $n_i$. Therefore, FedPFT can scale better compared to sending raw data or raw features when the number of samples is large. More specifically, when $n_i \gtrsim 2dCK$, it is more communication efficient to send $\mathcal{G}_{\text{diag}}(K)$ than send the raw features. This is also shown in Figure 6 (left). Similarly, equation (12) shows that the communication cost of $\mathcal{G}_{\text{spher}}(K = 1)$ is equal to the communication cost of sending the classifier head, which is $(Cd + C)$. Therefore, GMMs can have the same communication cost as FedAvg. Further, FedPFT supports heterogeneous communication resources, as each client can utilize a different $K$.

### 6.3 Evaluating against reconstruction attacks

We conduct reconstruction attacks on the various feature-sharing schemes described in this paper using the CIFAR10 dataset. Figure 7 (left) verifies the vulnerability of raw feature sharing – an attacker with access to in-distribution data (i.e. CIFAR-10 train set) can obtain high-fidelity reconstructions of the private data of clients (i.e. CIFAR-10 test set). This necessitates sharing schemes beyond sending raw features.

The attack we present involves training a U-Net-based conditional denoising diffusion model on (extracted feature, image) pairs and then performing inference on received feature embeddings. We apply the same attack on features sampled via FedPFT and DP-FedPFT. The reconstructions are *set-level*, and we present the closest image among the entire reconstruction set by SSIM. The resulting reconstructions do not resemble the original image (Figure 7 (left)). Quantitative results reporting image similarity metrics can be found in Table 7 (right).

For full experimental details, further quantitative results, and samples, please see Appendix H. We also include a full description of our threat model and results on different backbones (e.g. our attacks are stronger on MAEs), which could not be included in the main body due to space limitations.

### 6.4 Effect of data heterogeneity $\beta$

We analyze the effect of data heterogeneity on the performance of FedPFT and compare it to other one-shot baselines in Figure 8. We utilize CLIP, ViT-B/32 as our feature extractor and distribute the Caltech101 dataset across 50 clients according to Dirichlet with different values of $\beta$. Figure 8 confirms that FedPFT is invariant to data heterogeneity and outperforms other baselines by up to 32%. Further, it shows the challenge of model merging in a one-shot setting, since baselines such as averaging and FedBE do not perform as well as an ensemble with the decrease in data heterogeneity.

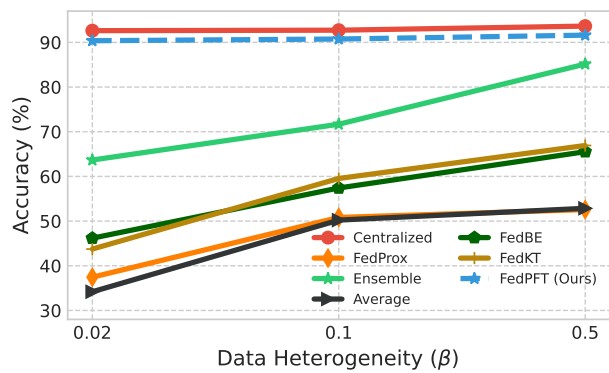

**Figure 8:** Effect of data heterogeneity ($\beta$) using the Caltech101 dataset with 50 clients. We use $K{=}1$ for FedPFT.

## 7 Conclusion

We introduce FedPFT, a one-shot federated learning (FL) method leveraging foundation models for better accuracy and communication efficiency. FedPFT utilizes per-client parametric models of features extracted from foundation models for data-free knowledge transfer. Our experiments show that FedPFT improves the communication-accuracy frontier in a wide range of data-heterogeneity settings. Moreover, we showed that FedPFT is amenable to formal privacy guarantees via differential privacy, exhibiting good privacy-accuracy tradeoffs. Our theoretical analysis demonstrates that FedPFT has server-side guarantees on the local accuracy of clients. Additionally, we conduct reconstruction attacks on feature-sharing schemes and demonstrate the privacy risks of sending real features.

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

# A Discussions and Limitations

## A.1 Why does FedPFT perform well with limited data?

The strong performance of FedPFT stems from its use of pre-trained foundational models to extract task-agnostic features. Additionally, GMMs require only a few parameters (centroids and variances) from these features, allowing them to perform well even with limited client data. Furthermore, since FedPFT recreates features at the server, it remains agnostic to data heterogeneity. Increasing GMM complexity—from spherical to diagonal and full covariance—enables better feature representation at the cost of higher communication overhead.

## A.2 Can we train any layer of the model using FedPFT?

We have explored training layers other than the classifier head using FedPFT. When training any layer of a network using FedPFT, we need to extract the outputs of the previous layer and learn a parametric model of those outputs. Two issues arise for layers that are not the classifier head: 1) the size of the output of hidden layers is very large and 2) simple parametric models cannot describe the output of hidden layers. For instance, consider using FedPFT for training the second to last layer (12th layer) of ViT-B/16 using FedPFT. In this case, we need to learn the outputs of the 11th layer which is a tensor of size (sequence=197, hidden-dim=768). In this case $d = 197 * 768 = 151,257$ is very large for both computation and transmission. More importantly, it is not clear how well GMMs model intermediate layer activations. We attribute the success of FedPFT to the observation that last layer outputs are well modeled by GMMs (See Figure 6). We believe the reason is that during the pre-training, last-layer activations of different classes were forced to be well-separated for classification using a linear probe. However, pre-last layer activations have positional information that does not aggregate well for calculating parametric statistics. We have tried modeling pre-last layers using GMMs but we were unsuccessful.

## A.3 Does sending label counts along with the GMMs to the server violate privacy?

Indeed, FedPFT sends label counts along with the GMMs to the server, which may not be advisable from a privacy perspective. However, this is not an inherent limitation of FedPFT. For example, one adaptation that can be made to address this is to employ *secure shufflers* (as used in private FL protocols, see (Kairouz et al., 2021) page 45). These are a class of cryptographic primitives that would shuffle the (GMM, label count) tuples provided by each client before the server receives them, and therefore, the server does not know the classes present on a specific client.

## A.4 Comparing FedPFT with model inversion methods

Model inversion methods like DENSE (Zhang et al., 2022) and DFRD (Wang et al., 2024) attempt to recreate the client's data at the server to distill the ensemble of the client's models to a global model using the generated synthetic data. Therefore, the global model would adhere to the performance of the ensemble of the client's models (Zhang et al., 2022). Based on the results of the ensemble presented in Sections 5 and 5.3, we argue that the performance of the ensemble model is also very limited due to the extreme data heterogeneity of clients. Further, these methods rely on training and transmitting the entire model and it is unclear how model inversion methods like DENSE can be extended to frozen, pre-trained foundational models as feature extractors. We were unsuccessful in our attempts to utilize DENSE for a pre-trained, frozen ResNet-50 to achieve competitive results. We believe one reason for our poor performance was the lack of access of DENSE to batch norm statistics of clients. Finally, inversion methods like DENSE and DFRD do not provide any privacy or accuracy guarantees.

## A.5 Using fine-tuned features vs zero-shot features

In section 5, we are reporting the centralized baseline for zero-shot features to represent the upper bound of what any method can achieve using a frozen, pre-trained feature extractor. In Table 6, we report the results when you can fine-tune a feature extractor. Table 6 demonstrates that fine-tuning hurts FedPFT performance,

and we attribute this to the fact that this introduces a mismatch between the client features computed with client feature extractors and the server feature extractor used at test time. For our baselines (with the exception being KD), fine-tuning and aggregating fine-tuned feature extractors improves performance. However, FedPFT with pre-trained feature extractors still outperforms them. Finally, an important caveat to fine-tuning and sharing models is the additional computational resources (FLOPS and memory) and communication costs, which become especially relevant when dealing with large foundation models. We will include this table along with the discussion in the final version.

### A.6 FedPFT for multi-shot FL

In this paper, our objective is to 1) bring and utilize large pre-trained foundation models in FL while at the same time 2) improve the communication cost by focusing only on one-shot FL. In the one-shot setting, FedPFT optimizes parametric distributions to capture feature representations. On the other hand, in the multi-round setting, it is not clear regarding the benefits: any approach that combines with feature fine-tuning, i.e., CCVR, will be bottlenecked by the communication of model weights of large foundation models. Further, the removal of the multi-round FL parts lets us prove formal accuracy and privacy guarantees.

### A.7 Comparing FedPFT with CCVR using a pre-trained model

CCVR, when using a pre-trained model, does not perform similarly to FedPFT (K=1). This is because, in FedPFT, we train the classifier head from *scratch* using Gaussian mixtures, whereas in CCVR, clients send a trained classifier head to the server, which then post-calibrates (*fine-tunes*) the classifier at the end of each round to mitigate bias and improve accuracy in multi-round FL. In Figure 4, we compare CCVR with a frozen pre-trained model against FedPFT, showing that FedPFT outperforms CCVR while also reducing communication costs by avoiding classifier head transmission. We believe that the bias in CCVR's local classifier heads prevents it from achieving competitive performance compared to FedPFT.

### A.8 Other limitations

One of the main limitations of our work is that we have not explored non-vision tasks. We will leave non-vision tasks for future work. Further, we leave other attacks including membership inference attacks for future work.

## B Societal Impact

This work promotes privacy by bringing foundation models to FL and using differential privacy. Unlike traditional approaches that require clients to send their raw data to a server for training, our method utilizes FL and differential privacy to allow users to send features instead. We show that clients do need to sacrifice privacy to achieve competitive performance. We further evaluate our method against privacy attacks and highlight the capability of private, feature-sharing methods for knowledge transfer using foundation models.

## C Algorithm

In this section, we provide the complete algorithm for FedPFT. Algorithm 1, describes the algorithm for FedPFT.

---

**Algorithm 1** FedPFT for centralized, one-shot FL.

1: **Input**: Client datasets $D_1, ..., D_I$, pre-trained feature extractor $f$.
2: **Parameters**: Number of clients $I$, number of classes $C$, number of mixtures $K$, covariance type cov.
3: **Output**: Model $w := h \circ f$

4: `// Client side:`
5: **for** each client $i \in \{1, ..., I\}$ **do**
6:     **for** each class $c \in \{1, ..., C\}$ **do**
7:         Let $F^{i,c} := \{f(\mathbf{x}) : (\mathbf{x}, y) \in D_i, y = c\}$.
8:         Run the EM algorithm on $F^{i,c}$ to learn a GMM $g^{i,c} \in \mathcal{G}_{\text{cov}}(K)$.
9:         Send $g^{i,c}$ parameters $\{(\pi_k^{i,c}, \boldsymbol{\mu}_k^{i,c}, \boldsymbol{\Sigma}_k^{i,c})\}_{k=1}^K$ to the server.
10:     **end for**
11: **end for**

12: `// Server side:`
13: **for** each received $\{(\pi_k^{i,c}, \boldsymbol{\mu}_k^{i,c}, \boldsymbol{\Sigma}_k^{i,c})\}_{k=1}^K$ set **do**
14:     Sample synthetic features $\tilde{F}^{i,c} \sim g^{i,c} = \sum_{k=1}^K \pi_k^{i,c} \cdot \mathcal{N}(\boldsymbol{\mu}_k^{i,c}, \boldsymbol{\Sigma}_k^{i,c})$ of size $|F^{i,c}|$.
15: **end for**
16: Let $\tilde{D} = \bigcup_{i=1}^I \bigcup_{c=1}^C \{(\mathbf{v}, c) : \mathbf{v} \in \tilde{F}^{i,c}\}$.
17: Train a classifier head $h$ on $\tilde{F}$, minimizing $\mathbb{E}_{(\mathbf{v},y) \sim \tilde{D}}[\ell(h; \mathbf{v}, y)]$ where $\ell$ is the cross-entropy loss.
18: **return** model $w = h \circ f$.

---

# D   Code

In this section, we provide code for the main component of FedPFT which is extracting the GMMs for class-conditional features. We consider the case of $k = 1$ and refer readers to the DP-EM (Park et al., 2017) method for the general case. Also, in this section, we assume each client can access only one class for simplicity and without losing generality.

```python
from sklearn.mixture import GaussianMixture as GMM
import numpy as np

def create_gmm(features: np.array, labels: np.array) -> Dict:
    '''
    Creates a diagonal GMM with 10 mixtures for each class in features and
    sample from it to create synthetic dataset

    Args:
        features(np.array): all the features of training dataset
        labels(np.array): all the associated labels of features
    returns 'synthetic_dataset'

    '''
    synthetic_dataset = {}
    for label in list(set(labels)):
        conditional_features = features[labels==label]

        # Create a GMM for 'label'
        gmm = GMM(
            n_components=10,
            covariance_type='diag',
            )
        gmm.fit(conditional_features)

        # Sample from the GMM for 'label' at the server
        gmm_feature, _ = gmm.sample(conditional_features.shape[0])
```

```
31
32          # Add it to the synth dict
33          synthetic_dataset[label] = gmm_feature
34
35      return synthetic_dataset
```

# E  DP-FedPFT

This section provides detailed definitions and proofs for Theorem 4.1.

**Definition E.1.** (Differential Privacy, (Dwork et al., 2006)). A randomized algorithm $\mathcal{M} : \mathcal{U} \to \Theta$ is $(\epsilon, \delta)$-differentially private if for every pair of neighboring datasets $D, D' \in \mathcal{U}$ for all $S \subseteq \Theta$, we have

$$\Pr[\mathcal{M}(D) \in S] \leq e^\epsilon \Pr[\mathcal{M}(D') \in S] + \delta. \tag{13}$$

**Lemma E.2.** *(Gaussian Mechanism, (Dwork et al., 2006; 2014)). Let $\epsilon > 0$ and let $g : D^n \to \mathbb{R}^d$ be a function with $\ell_2$-sensitivity $\Delta_g$. Then the Gaussian mechanism*

$$\mathcal{M}(D) := g(D) + \mathcal{N}\left(0, \left(\frac{\Delta_g \sqrt{2ln(2/\delta)}}{\epsilon}\right)^2 \cdot \mathbb{I}_{d \times d}\right), \tag{14}$$

*satisfies $(\epsilon, \delta)$-differential privacy.*

*Proof.* (Theorem 4.1) We wish to apply Lemma E.2 with $g$ mapping a feature dataset to its Gaussian mixture approximation obtained via EM-method. The post-processing property (Proposition 2.1 of (Dwork et al., 2014)) ensures that the projection onto the positive semi-definite symmetric matrices preserves differential privacy. For $k = 1$, denote by $\hat{\boldsymbol{\mu}}, \hat{\boldsymbol{\Sigma}}$ the mean and covariance matrix of this Gaussian approximation; in this instance they coincide with the mean and covariance matrix of the dataset. We may use the $\ell_2$-sensitivity of the function $(\hat{\boldsymbol{\mu}}, \hat{\boldsymbol{\Sigma}})$ to apply the aforementioned Lemma.

To begin with, for any triplet of independent random variables $(X, Y, \epsilon)$ with $X, Y$ with values in $d \times 1$ matrices with real entries and $\epsilon$ in $\{0, 1\}$ with $\mathbb{P}(\epsilon = 1) = p$, denote $X^\epsilon := \epsilon X + (1-\epsilon)Y$ the mixture of $X$ and $Y$:

$$\mathbb{E}(X^\epsilon) = \mathbb{E}(X) + (1-p)(\mathbb{E}(Y) - \mathbb{E}(X)); \tag{15}$$

$$\mathrm{Cov}\left(X^\epsilon, X^\epsilon\right) = \mathrm{Cov}\left(X, X\right) + (1-p)\left[\mathrm{Cov}(Y, Y) - \mathrm{Cov}(X, X)\right] + p(1-p)(\mathbb{E}(X) - \mathbb{E}(Y))(\mathbb{E}(X) - \mathbb{E}(Y))^T. \tag{16}$$

If $X$ follows the uniform distribution on a dataset $D'$ with $n-1$ element, $Y$ is deterministic with value at some $x_n \in \mathbb{R}^d$ and $p = 1 - 1/n$ ; then $X^\epsilon$ follows the uniform distribution on the dataset $D'' = D' \cup \{x_n\}$. Assuming $D'$ and $D''$ are in the ball of radius 1, we have:

$$\|\mathbb{E}(X^\epsilon) - \mathbb{E}(X)\|_2 = (1-p)\|(\mathbb{E}(Y) - \mathbb{E}(X))\|_2 \leq \frac{2}{n};$$

$$\begin{aligned}
\|\mathrm{Cov}\left(X^\epsilon, X^\epsilon\right) - \mathrm{Cov}\left(X, X\right)\|_F^2 &= (1-p)^2 \|\mathrm{Cov}(X, X)\|_F^2 \\
&\quad + p^2(1-p)^2 \mathrm{Tr}\left(\mathbb{E}(X) - \mathbb{E}(Y))(\mathbb{E}(X) - \mathbb{E}(Y))^T(\mathbb{E}(X) - \mathbb{E}(Y))(\mathbb{E}(X) - \mathbb{E}(Y))^T\right); \\
&\quad - 2p(1-p)^2 \mathrm{Tr}\left(\mathrm{Cov}(X, X)(\mathbb{E}(X) - \mathbb{E}(Y))(\mathbb{E}(X) - \mathbb{E}(Y))^T\right) \\
&= (1-p)^2 \|\mathrm{Cov}(X, X)\|_F^2 + p^2(1-p)^2 \|\mathbb{E}(X) - \mathbb{E}(Y)\|_2^4 \\
&\quad + 2p(1-p)^2 (\mathbb{E}(X) - \mathbb{E}(Y))^T \mathrm{Cov}(X, X)(\mathbb{E}(X) - \mathbb{E}(Y)) \\
&\leq \frac{1}{n^2} \|\hat{\boldsymbol{\Sigma}}\|_F^2 + 16\frac{(n-1)^2}{n^4} + \frac{8(n-1)}{n^3}\rho(\hat{\boldsymbol{\Sigma}}) \\
&\leq \frac{1}{n^2}\left(4 + 16 + 8 \times 2\right).
\end{aligned}$$

Taking the square root of both sides we have:

$$\|\mathrm{Cov}\left(X^{\epsilon}, X^{\epsilon}\right) - \mathrm{Cov}\left(X, X\right)\|_F \leq \frac{6}{n},$$

where $\|\cdot\|_F$ is the Frobenius norm, $\rho$ return the largest eigenvalue. We used the bounds $\rho(\hat{\boldsymbol{\Sigma}}) \leq \|\hat{\boldsymbol{\Sigma}}\|_F$ and $\|\hat{\boldsymbol{\Sigma}}\|_F \leq 2$; the latter may be obtained in a similar fashion by developing $\mathrm{Cov}(X, X)$ using $X = \sum_{i=1}^{n-1} \eta_i X_i$ where each $X_i$ is deterministic at some $x_i$ and $\eta$ is uniform on the set of binary vectors of length $n-1$ satisfying $\sum_i \eta_i = 1$.

Finally, the $\ell_2$-sensitivity of $(\hat{\boldsymbol{\mu}}, \hat{\boldsymbol{\Sigma}})$ is $\sqrt{\left(\frac{2}{n}\right)^2 + \left(\frac{6}{n}\right)^2} = \frac{2\sqrt{10}}{n}$. Inserting the $\ell_2$-sensitivity in equation (14) yields the result. $\qquad\square$

*Remark* E.3. Note that in (Park et al., 2017), they derive an $\ell_2$-sensitivity bound for $\hat{\boldsymbol{\Sigma}}$ of $2/n$ instead of our $6/n$. The reason is that one may replace $(\hat{\boldsymbol{\mu}}, \hat{\boldsymbol{\Sigma}})$ by $(\hat{\boldsymbol{\mu}}, \hat{\boldsymbol{\Sigma}} + \hat{\boldsymbol{\mu}}\hat{\boldsymbol{\mu}}^T)$ to reduce the $\ell_2$-sensitivity of the covariance part. This leads to the improved total $\ell_2$-sensitivity of $\frac{2\sqrt{2}}{n}$.

On the server side, the Gaussian mechanism would return $(\hat{\boldsymbol{\mu}} + \Delta\boldsymbol{\mu}, \hat{\boldsymbol{\Sigma}} - \hat{\boldsymbol{\mu}}\hat{\boldsymbol{\mu}}^T + \Delta\boldsymbol{\Sigma})$. The server would thus reconstruct $\hat{\boldsymbol{\Sigma}}$ by computing

$$\begin{aligned}
\hat{\boldsymbol{\Sigma}}_{\mathrm{server}} &= \hat{\boldsymbol{\Sigma}} - \hat{\boldsymbol{\mu}}\hat{\boldsymbol{\mu}}^T + \Delta\boldsymbol{\Sigma} + (\hat{\boldsymbol{\mu}} + \Delta\boldsymbol{\mu})(\hat{\boldsymbol{\mu}} + \Delta\boldsymbol{\mu})^T \\
&= \hat{\boldsymbol{\Sigma}} + \hat{\boldsymbol{\mu}}\Delta\boldsymbol{\mu}^T + \Delta\boldsymbol{\mu}\hat{\boldsymbol{\mu}}^T + \Delta\boldsymbol{\mu}\Delta\boldsymbol{\mu}^T + \Delta\boldsymbol{\Sigma}.
\end{aligned}$$

Therefore, for a given coefficient $(i, j)$ of the reconstructed covariance matrix $\hat{\boldsymbol{\Sigma}}_{\mathrm{server}}$, the error term is $\hat{\mu}_i \Delta\mu_j + \hat{\mu}_j \Delta\mu_i + \Delta\mu_i \Delta\mu_j + \Delta\Sigma_{ij}$. Assuming a Gaussian noise of standard deviation $\sigma = \frac{2\sqrt{2}}{n}C$ with $C = \frac{\sqrt{2\ln(2/\delta)}}{\epsilon}$ for both $\Delta\boldsymbol{\Sigma}$ and $\Delta\boldsymbol{\mu}$ the standard deviation of the reconstruction error is then

$$\sigma_{\mathrm{reconstruction}} = \sqrt{(\hat{\mu}_i^2 + \hat{\mu}_j^2)\sigma^2 + \sigma^4 + \sigma^2} \leq \sigma\sqrt{2 + \sigma^2} \leq \frac{2\sqrt{2}C\sqrt{2 + C^2/n^2}}{n}.$$

Compared to the reconstruction error of the Gaussian mechanism we used in our experiments $\sigma'_{\mathrm{reconstruction}} = \frac{2\sqrt{10}}{n}C$, the more sophisticated methods we described above may allow better reconstruction error without sacrificing the differential privacy for big enough datasets, ie if $n \geq \frac{\sqrt{2\ln(2/\delta)}}{\sqrt{3}\epsilon}$.

Another consequence is that the noises on different elements of the covariance matrix are not independent.

*Remark* E.4. The assumption of normalized features, i.e., $\|f(\mathbf{x})\|_2 \leq 1$, in Theorem 4.1 does not limit the performance of networks with soft-max loss function since they both have the same expressive power as shown in Proposition 3.A in (Zhang et al., 2023).

# F   Control over 0-1 loss

Since the projection of from true feature distributions to mixture of Gaussian is lossy, one may control the consequence on the accuracy of the classifier: "Given a classifier $h : \mathcal{X} \to \{1, \cdots, C\}$ trained on a synthetic dataset, what guarantees do we have on the accuracy of $h$ used to classify the true dataset?".

We adress this question by proving a theoretical bound that my be rewritten using the loss of the local client training and the accuracy of the server model.

## F.1   A Theoretical bound

The setting may be formalized as follows:

- a finite set of classes $\mathcal{C} = \{1, \cdots, C\}$, a feature space $\mathcal{X}$ and an approximate feature space $\mathcal{Y}$;

- a true feature/label distribution $\alpha$ on $\mathcal{C} \times \mathcal{X}$ and an approximate feature/label distribution $\beta$ on $\mathcal{C} \times \mathcal{Y}$;

- a classifier $h_\alpha : \mathcal{X} \to \mathcal{C}$ trained on $\alpha$ and a classifier $h_\beta : \mathcal{Y} \to \mathcal{C}$ trained on $\beta$.

- an feature approximator mapping $\iota : \mathcal{X} \to \mathcal{Y}$

We denote by $\mathrm{Acc}(h, \alpha)$ the accuracy of the predictor $h$ evaluated using the distribution $\alpha$.

We prove a bound adapted to the purpose of linking the EM loss to the accuracy.

**Theorem F.1.** *Let $\alpha, \beta$ be probability distributions on $\mathcal{X} \times \mathcal{C}$ and $\mathcal{Y} \times \mathcal{C}$ with same marginal $\eta$ on $\mathcal{C}$. Let $h : \mathcal{Y} \to \mathcal{C}$ be any predictor and let $\iota : \mathcal{X} \to \mathcal{Y}$ be any measurable map. We have*

$$\mathrm{Acc}(h \circ \iota; \alpha) \geq \mathbb{E}_c \left[ \mathrm{Acc}(h; \beta|c) \times (\mathrm{Acc}(h; \beta|c) - \mathrm{div}_{TV}(\iota \# (\alpha|c), \beta|c)) \right]$$

*Proof.* Consider any coupling $\mathbb{P}$ of $\alpha$ and $\beta$ over $\mathcal{C}$ ie a distribution on $\mathcal{C} \times \mathcal{X} \times \mathcal{Y}$ whose marginals on $\mathcal{C} \times \mathcal{X}$ and $\mathcal{C} \times \mathcal{Y}$ are $\alpha$ and $\beta$ respectively[1].

Given some $c$,

$$\mathrm{Acc}(h \circ \iota; \alpha|c) = (\alpha|c)(\{x \; : \; h \circ \iota(x) = c\}) \tag{17}$$
$$= (\mathbb{P}|c)(\{(x,y) \; : \; h \circ \iota(x) = c\}) \tag{18}$$
$$\geq (\mathbb{P}|c)(\{(x,y) \; : \; h \circ \iota(x) = c \ \text{ and } \ \iota(x) = y\}) \tag{19}$$
$$= (\mathbb{P}|c)(\{(x,y) \; : \; h(y) = c\}) \times (\mathbb{P}|c, h(y) = c)(\{(x,y) \; : \; \iota(x) = y\}) \tag{20}$$
$$= \underbrace{(\beta|c)(\{y \; : \; h(y) = c\})}_{\mathrm{Acc}(h;\beta|c)} \times (\mathbb{P}|c, h(y) = c)(\{(x,y) \; : \; \iota(x) = y\}) \tag{21}$$

Since neither the left hand side of the first line nor the first term in the right hand side of the last line depend on the coupling $\mathbb{P}$ we chose, we may choose it to maximize $(\mathbb{P}|c, h(y) = c)(\{(x,y) \; : \; \iota(x) = y\}) = 1 - \mathbb{E}(\mathbf{1}_{Z \neq Y})$ with $Y$ drawn from $(\beta|c, h(y) = c)$ and $Z$ drawn from $\iota \#(\alpha|c)$. We recognize a characterization of the total variation as the minimal value of $\mathbb{E}(\mathbf{1}_{Z \neq Y})$ over couplings of $Z$ and $Y$. Then:

$$(\mathbb{P}|c, h(y) = c)(\{(x,y) \; : \; \iota(x) = y\}) \geq 1 - \mathrm{div}_{TV}(\iota \#(\alpha|c) \; ; \; (\beta|c, h(y) = c)) \tag{22}$$
$$\geq 1 - \mathrm{div}_{TV}(\iota \#(\alpha|c) \; ; \; (\beta|c)) - \mathrm{div}_{TV}((\beta|c) \; ; \; (\beta|c, h(y) = c)) \tag{23}$$
$$= 1 - \mathrm{div}_{TV}(\iota \#(\alpha|c) \; ; \; (\beta|c)) - (1 - \mathrm{Acc}(h(y); \beta|c)) \tag{24}$$
$$= \mathrm{Acc}(h(y); \beta|c) - \mathrm{div}_{TV}(\iota \#(\alpha|c) \; ; \; (\beta|c)). \tag{25}$$

We use triangular inequality of total variation to get the second line. The third line is obtained by applying the general property that for any probability distribution $\mathbb{Q}$ and any event $E$ with $\mathbb{Q}(E) > 0$ we have

$$\mathrm{div}_{TV}(\mathbb{Q}, (\mathbb{Q}|E)) = \mathbb{Q}(\overline{E}).$$

We thus have

$$\mathrm{Acc}(h \circ \iota; \alpha|c) \geq \mathrm{Acc}(h; \beta|c) \times [\mathrm{Acc}(h(y); \beta|c) - \mathrm{div}_{TV}(\iota \#(\alpha|c) \; ; \; (\beta|c))]$$

The result follows by taking expectation over $c$ distributed along $\eta$. $\qquad \square$

## F.2   Application to FedPFT

We now switch back to the notations used in the preliminaries. Using Pinsker inequality (Csiszár & Körner, 2011), we obtain the following bound for the accuracy of the server model

$$\mathrm{Acc}(h, F^i) \geq \mathbb{E}_c \left[ \mathrm{Acc}(h, \widetilde{F}^{i,c}) \times \left( \mathrm{Acc}(h, \widetilde{F}^{i,c}) - \sqrt{\frac{1}{2} \left( \mathrm{div}_{KL}(\widetilde{F}^{i,c} || F^{i,c}) \right)} \right) \right]. \tag{26}$$

---

[1]We define the conditioning of a probability distribution $\mathbb{P}$ by an event $E$ having $\mathbb{P}(E) > 0$ is the distribution $\frac{\mathbf{1}_E \mathbb{P}}{\mathbb{P}(E)}$ so that it is still a distribution on the same underlying measurable space.

Since the EM algorithm maximizes the log likelyhood of the Gaussian mixture given dataset samples, we may insert $\operatorname{div}_{KL}(\widetilde{F}^{i,c}||F^{i,c}) = \mathcal{H}^{i,c} - \mathcal{L}_{EM}^{i,c}$ to obtain

$$\operatorname{Acc}(h, F^i) \geq \mathbb{E}_c \left[ \operatorname{Acc}(h, \widetilde{F}^{i,c}) \times \left( \operatorname{Acc}(h, \widetilde{F}^{i,c}) - \sqrt{\frac{1}{2} \left( \mathcal{H}^{i,c} - \mathcal{L}_{EM}^{i,c} \right)} \right) \right] \tag{27}$$

where $\mathcal{H}^{i,c}$ is the self-entropy of the distribution $F^{i,c}$ of features with label $c$ in client $i$ and $\mathcal{L}_{EM}^{i,c}$ is the log-likelyhood of the Gaussian mixture $\widetilde{F}^{i,c}$ trained using the EM algorithm. The last equation yields inequation 9 by replacing accuracies by 0-1 losses. Beware that the feature distribution is a priori discrete, in order to evaluate the entropy we need to dequantize the dataset, otherwise, $\mathcal{H}^{i,c} = +\infty$ and the bound is useless.

### F.3 Comparison to earlier bounds

Commonly known bounds on the accuracy of $h$ may be found in (Ben-David et al., 2010), but these bounds are unfortunately not directly useful in our setting. Indeed, they depend on an estimation on how much server/client class predictors differ.

More precisely, in the limit of perfect accuracy of the server class predictor, ie $\forall i, c, \ \operatorname{Acc}(h, \widetilde{F}^{i,c}) \simeq 1$, our bound yields

$$\operatorname{Acc}(h, F^i) \gtrsim \mathbb{E}_c \left[ 1 - \sqrt{\frac{1}{2} \left( \mathcal{H}^{i,c} - \mathcal{L}_{EM}^{i,c} \right)} \right]. \tag{28}$$

In this limit, the bound deduced from that of (Ben-David et al., 2010) contains an additional intractable negative term on the right hand side:

$$\operatorname{Acc}(h, F^i) \geq \mathbb{E}_c \left[ 1 - \sqrt{\frac{1}{2} \left( \mathcal{H}^{i,c} - \mathcal{L}_{EM}^{i,c} \right)} \right] - \mathbb{E}_c \min \left( \mathbb{E}_{x \sim F^{i,c}} \left[ \mathbf{1}_{h(x) \neq h_{\text{client}}^*(x)} \right] ; \mathbb{E}_{x \sim \widetilde{F}^{i,c}} \mathbf{1}_{h(x) \neq h_{\text{client}}^*(x)} \right) \tag{29}$$

where $h_{\text{client}}^*$ is a hypothetical class predictor perfectly fine-tuned by the client with its own data $f(D_i)$. However, our bound may be less sharp in general as the positive term is in $\operatorname{Acc}(h, \widetilde{F}^{i,c})^2$. This leads to a quick degradation of the theoretical accuracy guarantee as the measured server accuracy drops. In general, we should have error on the server side at least twice lesser than the target error on the client side in order to hope for achievable EM-loss target on client side.

## G   Experiments

In this section, we provide the full details of all of our experiments and datasets.

### G.1   Implementation details

We use the Pytorch library for the implementation of our methods. We use a cluster of 4 NVIDIA v100 GPUs for our experiments. All of our experiments can be run on a single v100 GPU.

**Datasets.**   We provide the complete details of the dataset we used in this paper in Table 4.

**Learning rates and optimizers.**   We use Adam with a learning rate of $1e^{-4}$ for training the classifier head for FedPFT, Ensemble, FedFT, and centralized training.

**Knowledge Distillation.**   For the implementation of KD methods, we use Adam for training the local classifier heads with $1e^{-4}$ learning rate. We locally train the models for 100 epochs and keep the best model on the local test dataset. Then, we distill from the source to the destination model in 50 epochs with the same optimizer and classifier head. We test three temperature values $\{1, 5, 10\}$ and report the best results

**FedAvg.**   We test three local training epochs $\{50, 100, 200\}$ and three learning rates $\{5e^{-2}, 1e^{-2}, 1e^{-2}\}$ and report the best result.

**FedYogi.**   We set the server learning rate $\eta$ to 0.01, $\beta_1$, $\beta_2$, and $\tau$ to 0.9, 0.99, and 0.001 respectively. We also initialize vector $v_t$ to $1e^{-6}$. For the rest of the hyperparameters, including the clients' learning rates, we use the same hyperparameters as FedAvg.

**FedProx.**   We add a regularizer with a weight of 0.01 to the loss function of each client during local training to penalize divergence. We use the same hyperparameters as FedAvg for the rest of the hyperparameters.

**DSFL.**   We set the top-K sparsification ($K$ in the DSFL paper) as half of the number of parameters of the classifier head.

**FedBE.**   We use Adam with a learning rate of $1e^{-4}$ for training the classifier head and we sample 15 models from the posterior distribution of classifier heads.

**CCVR.**   For CCVR, we use $M_C = 100$ as stated in (Luo et al., 2021). We also utilize a diagonal covariance matrix to make the algorithm competitive in terms of communication cost. The rest of the method is similar to FedAvg.

**Table 4:** Summary of datasets

| Dataset | Image size | # Train | # Testing | #Classes | Feature extractor |
|---|---|---|---|---|---|
| CIFAR10 | (32, 32) | 50,000 | 10,000 | 10 | ResNet-50 |
| CIFAR100 | (32, 32) | 50,000 | 10,000 | 100 | ResNet-50 |
| PACS (P) | (224, 224) | 1,336 | 334 | 7 | Base ViT, 16 |
| PACS (S) | (224, 224) | 3144 | 785 | 7 | Base ViT, 16 |
| Office Home (C) | min:(18, 18) | 3492 | 873 | 65 | Base ViT, 16 |
| Office Home (P) | min:(75, 63) | 3551 | 888 | 65 | Base ViT, 16 |
| Caltech101 | min:(200, 200) | 6084 | 3060 | 101 | CLIP, ViT-B/32 |
| Stanford Cars | min:(360, 240) | 12948 | 3237 | 196 | CLIP, ViT-B/32 |
| Oxford Pets | min:(108, 114) | 3680 | 3669 | 37 | CLIP, ViT-B/32 |
| Food101 | max:(512, 512) | 75750 | 25250 | 101 | CLIP, ViT-B/32 |

### G.2 Comparing to existing one-shot and multi-shot FL methods

In this section, we provide the details for the experiment in Section 5.2. Table 5 summarizes one-shot methods results in Table format for Figure 4. Further, Figure 10 and 9 shows the data partitions based on Dirichlet shift with $\beta = 0.1$ with 50 clients for Caltech101 and CIFAR100 Datasets, respectively. The magnitude of data samples for each class label in each client is represented by the size of the red circle. This figure shows the non-iidness of data distribution among clients.

**Table 5:** one-shot methods results in Table format for Figure 4

| Methods | CIFAR100 ($\beta = 0.1$) Accuracy | Comm. | Caltech101 ($\beta = 0.1$) Accuracy | Comm. |
|---|---|---|---|---|
| Centralized | $73.50 \pm 0.04$ | 97 MB | $95.36 \pm 0.08$ | 3.6 MB |
| Ensemble | $53.97 \pm 0.49$ | 19 MB | $72.86 \pm 2.09$ | 4.9 MB |
| AVG | $43.51 \pm 0.26$ | 19 MB | $56.90 \pm 0.67$ | 4.9 MB |
| FedKT | $51.80 \pm 0.02$ | 19 MB | $46.64 \pm 0.33$ | 4.9 MB |
| FedBE | $56.32 \pm 0.44$ | 19 MB | $57.67 \pm 0.27$ | 4.9 MB |
| $\hat{\mathcal{G}}_{\mathrm{spher}}(k=1)$ | $56.96 \pm 0.23$ | 22 MB | $86.41 \pm 0.13$ | 3 MB |
| $\mathcal{G}_{\mathrm{diag}}(k=10)$ | $69.91 \pm 0.11$ | 0.3 GB | $94.67 \pm 0.03$ | 23 MB |
| $\mathcal{G}_{\mathrm{diag}}(k=50)$ | $72.05 \pm 0.10$ | 0.9 GB | $94.74 \pm 0.13$ | 33 MB |
| $\mathcal{G}_{\mathrm{diag}}(k=100)$ | $\mathbf{72.29 \pm 0.08}$ | 1.2 GB | $\mathbf{94.83 \pm 0.18}$ | 36 MB |
| $\mathcal{G}_{\mathrm{spher}}(k=1)$ | $60.16 \pm 0.04$ | 22 MB | $93.46 \pm 0.16$ | 3 MB |
| $\mathcal{G}_{\mathrm{spher}}(k=10)$ | $69.74 \pm 0.16$ | 0.2 GB | $94.59 \pm 0.13$ | 12 MB |
| $\mathcal{G}_{\mathrm{spher}}(k=50)$ | $\mathbf{71.88 \pm 0.03}$ | 0.5 GB | $\mathbf{94.71 \pm 0.16}$ | 16 MB |

### G.3 Effect of fine-tuning feature extractor

In this section, we investigate the effect of fine-tuning the feature extractor. More specifically, in Table 6, which is an extension of label shift setup in Table 3, we also fine-tune the feature extractor as well as the classifier head. From Table 6, We see that fine-tuning hurts FedPFT performance, and we attribute

**Table 6:** Knowledge transfer results for label-shift

| Methods | CIFAR-10 Accuracy | Comm. | CIFAR-100 Accuracy | Comm. |
|---|---|---|---|---|
| Centralized (fine-tuned) | $97.33 \pm 0.09$ | 71.6 MB | $85.18 \pm 0.08$ | 71.6 MB |
| Centralized (frozen) | $90.85 \pm 0.03$ | 97.6 MB | $73.97 \pm 0.06$ | 97.6 MB |
| Ensemble (fine-tuned) | $85.23 \pm 0.01$ | 89.6 MB | $57.94 \pm 0.22$ | 89.7 MB |
| Ensemble (frozen) | $80.18 \pm 0.30$ | 80.0 KB | $74.78 \pm 0.57$ | 0.78 MB |
| Average (fine-tuned) | $83.18 \pm 2.63$ | 89.6 MB | $68.63 \pm 1.06$ | 89.7 MB |
| Average (frozen) | $77.66 \pm 1.04$ | 80.0 KB | $56.82 \pm 0.22$ | 0.78 MB |
| KD (fine-tuned) | $58.79 \pm 3.94$ | 89.6 MB | $44.38 \pm 0.26$ | 89.7 MB |
| KD (frozen) | $74.22 \pm 0.42$ | 80.0 KB | $55.62 \pm 0.20$ | 0.78 MB |
| FedPFT (fine-tuned, K=20) | $76.23 \pm 0.50$ | 0.8 MB | $60.46 \pm 0.03$ | 7.8 MB |
| FedPFT (frozen, K=20) | $86.89 \pm 0.02$ | 0.8 MB | $70.31 \pm 0.10$ | 7.8 MB |

this to the fact that this introduces a mismatch between the client features computed with client feature extractors and the server feature extractor used at test time. For our baselines (with the exception being KD), fine-tuning and aggregating feature extractors improves performance. However, FedPFT with pre-trained feature extractors still outperforms them.

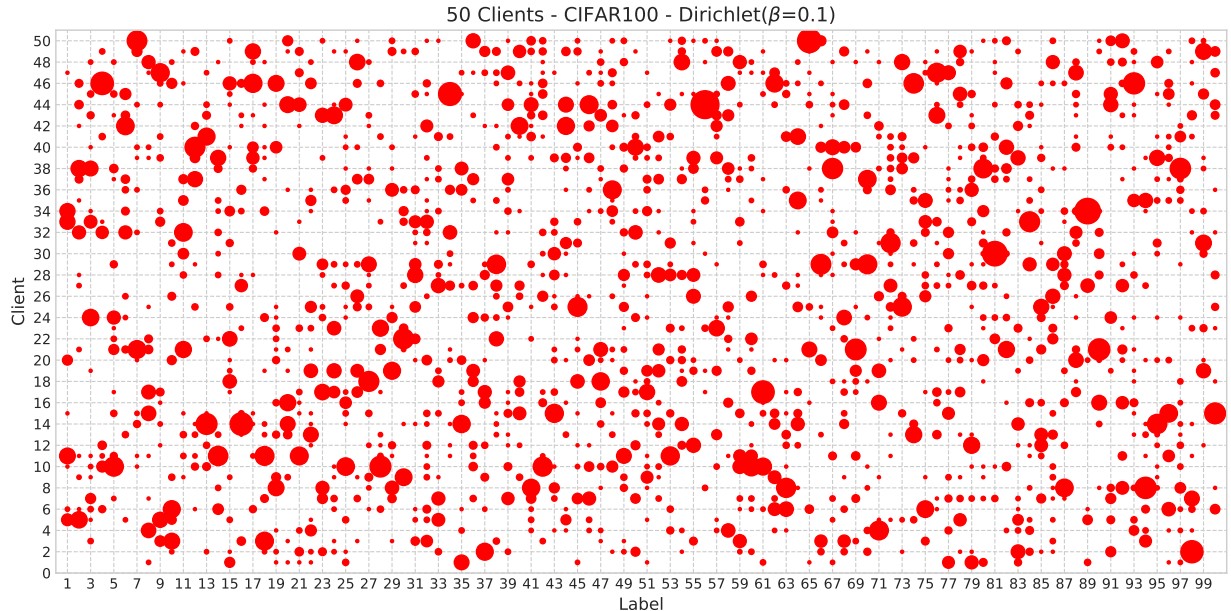

**Figure 9:** Data partitions based on Dirichlet shift with $\beta = 0.1$ with 50 clients for CIFAR100 datasets. The size of the red circle represents the magnitude of data samples for each class label in each client

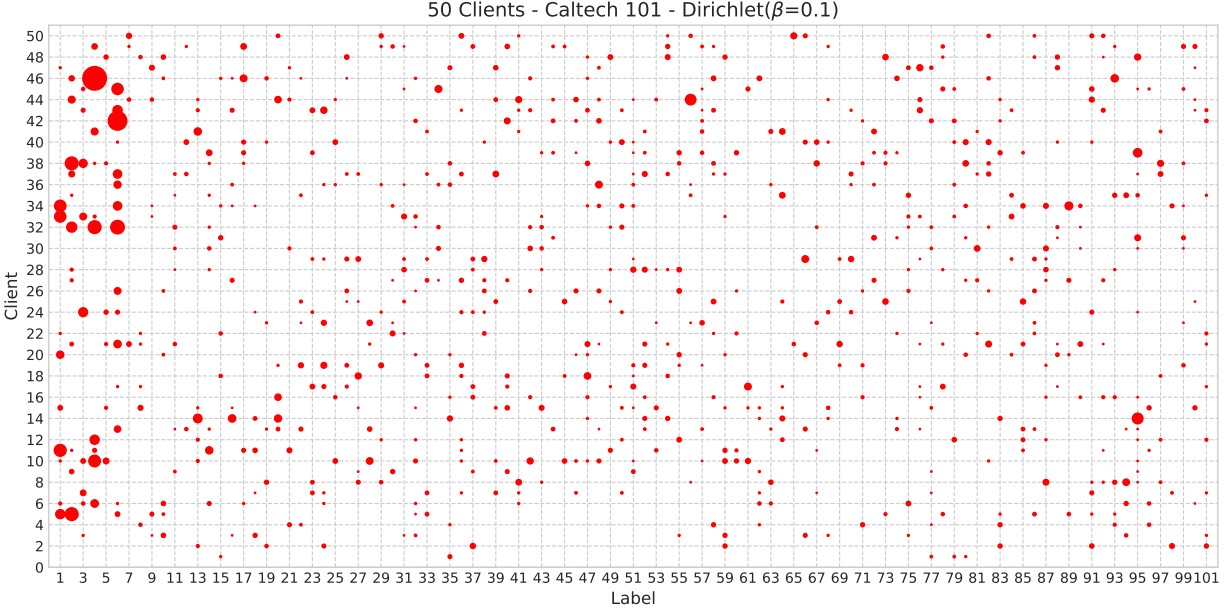

**Figure 10:** Data partitions based on Dirichlet shift with $\beta = 0.1$ with 50 clients for Caltech101 datasets. The magnitude of data samples for each class label in each client is represented by the size of the red circle

### G.4 Details of experiments for peer-to-peer experiments

In this section, we provide the full details of experiments in Section 5.3 and Table 3. Table 7 is the complete version of Table 3 where we also include the communication cost of each method.

For baselines in Table 3, we compare to KD, where: (1) each client locally trains a classifier head; (2) the source client sends its local classifier head to the destination client; and (3) the destination client distills

**Table 7:** FedPFT in three extreme shifts in two-client decentralized FL.

| | Disjoint Label shift | | | | Covariate shift | | | | Task shift | | | |
| | CIFAR-10 | | CIFAR-100 | | PACS (P→S) | | Office Home (C→P) | | Birds → Cars | | Pets → Food | |
| Methods | Accuracy | Comm. | Accuracy | Comm. | Accuracy | Comm. | Accuracy | Comm. | Accuracy | Comm. | Accuracy | Comm. |
|---|---|---|---|---|---|---|---|---|---|---|---|---|
| Centralized | $90.85$ $_{\pm 0.03}$ | 97 MB | $73.97$ $_{\pm 0.06}$ | 0.4 97 MB | $89.15$ $_{\pm 0.17}$ | 2.5 MB | $82.00$ $_{\pm 0.16}$ | 6.3 MB | $81.88$ $_{\pm 0.06}$ | 6.7 MB | $88.48$ $_{\pm 0.05}$ | 3.6 MB |
| Ensemble | $80.18$ $_{\pm 0.30}$ | 80 KB | $57.94$ $_{\pm 0.22}$ | 0.7 MB | $79.59$ $_{\pm 0.83}$ | 10.5 KB | $71.36$ $_{\pm 0.46}$ | 96 KB | $58.33$ $_{\pm 2.01}$ | 0.3 MB | $83.54$ $_{\pm 0.21}$ | 0.2 MB |
| Average | $77.66$ $_{\pm 1.04}$ | 80 KB | $56.82$ $_{\pm 0.22}$ | 0.7 MB | $77.83$ $_{\pm 0.23}$ | 10.5 KB | $69.69$ $_{\pm 0.69}$ | 96 KB | $72.65$ $_{\pm 0.13}$ | 0.3 MB | $83.25$ $_{\pm 0.59}$ | 0.2 MB |
| KD | $74.22$ $_{\pm 0.42}$ | 80 KB | $55.62$ $_{\pm 0.20}$ | 0.7 MB | $67.69$ $_{\pm 2.47}$ | 10.5 KB | $72.15$ $_{\pm 0.75}$ | 96 KB | $40.46$ $_{\pm 0.31}$ | 0.3 MB | $43.67$ $_{\pm 0.04}$ | 0.2 MB |
| $\mathcal{G}_{\text{diag}}$(K=10) | $86.19$ $_{\pm 0.15}$ | 0.4 MB | $69.97$ $_{\pm 0.07}$ | 3.9 MB | **89.12** $_{\pm 0.18}$ | 0.2 MB | $80.56$ $_{\pm 0.31}$ | 1.8 MB | $81.74$ $_{\pm 0.06}$ | 1.9 MB | $88.22$ $_{\pm 0.07}$ | 0.7 MB |
| $\mathcal{G}_{\text{diag}}$(K=20) | **86.89** $_{\pm 0.02}$ | 0.8 MB | **70.31** $_{\pm 0.10}$ | 7.8 MB | $89.00$ $_{\pm 0.15}$ | 0.4 MB | **80.94** $_{\pm 0.16}$ | 3.6 MB | **81.75** $_{\pm 0.04}$ | 3.8 MB | **88.25** $_{\pm 0.08}$ | 1.4 MB |

the received classifier head to its local classifier head. We report ensembling and averaging locally trained classifier heads as baselines.

In our **disjoint label** shift setups, our source client only has samples from the first half of labels (0-4 for CIFAR-10 or 0-49 for CIFAR-100), and the destination client has the other half. In our **covariate shift** setups, the two clients have access to the two most distinctive domains of PACS and Office Home datasets according to (Hemati et al., 2023). Specifically, for PACS, we consider the scenario where the source has access only to Photo (P) images, while the destination has access to Sketch (S) images. For Office Home, the source has access to Clipart (C) images, and the destination has access to Product (P) images. In our **task shift** setups, the two clients have access to two different datasets with distinct tasks. First, we consider the scenario where the source has bird images from Caltech101, while the destination has car images from Stanford Cars. In the second experiment, the source has access to pet images from Oxford Pets, and the destination has food images from Food101.

### G.5 Effect of $\epsilon$

In this section, we evaluate the effect of $\epsilon$ using the CIFAR-10 settings from the experiment in Table 3. As shown in Figure 11, increasing $\epsilon$ leads the performance of DP-FedPFT to approach that of FedPFT.

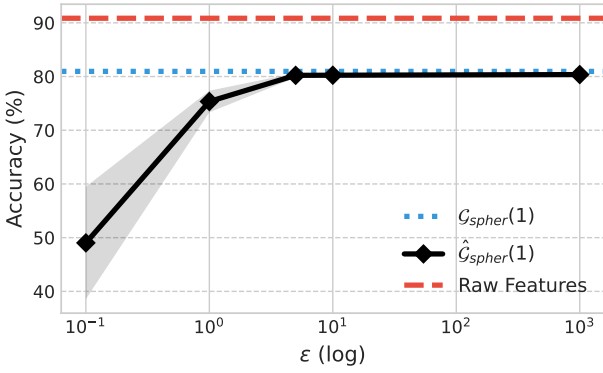

**Figure 11:** Effect of $\epsilon$ on the performance of DP-FedPFT

## H   Reconstruction attack details

We conduct reconstruction attacks on the various feature-sharing schemes described in this paper. First, we verify the vulnerability of raw feature sharing: we demonstrate that an attacker with access to in-distribution data can obtain high-fidelity reconstructions of private training data. This necessitates sharing schemes beyond sending raw features. The attack we present proceeds by training a generative model on (feature, image) pairs, and then performing inference on received feature embeddings. Next, we apply the aforementioned attack on features sampled from GMMs received via our proposed scheme. We find that the resulting reconstructions do not resemble real training data.

### H.1 Threat model

We consider the setting where two clients (a *defender* and an *attacker*) collaborate via a feature-sharing scheme to train a model to perform well on the union of their datasets. *Both parties* have:

- Local datasets of (image, label) pairs, which we call $D_d$ and $D_a$ respectively, where $D_* = \{(x_i, y_i)\}_{i=1}^{m_*}$.
- Black-box access to a feature embedding function $E : \mathcal{X} \to \mathcal{Z}$.

*The defender* produces an embedded version of their dataset $E(D_d) := \{(E(x_i), y_i)\}_{i=1}^{m_d}$, and passes it along to the attacker via a feature-sharing scheme. We consider sending: the embedded dataset $E(D_d)$ directly (*raw features*), our proposedfeature-sharing scheme (*FedPFT*), as well as with differentially privacy (*DP-FedPFT*).

*The attacker* has a dataset $D_a$, which is assumed to be in-distribution of the defender's dataset $D_d$. Concretely: $D_a$ is the CIFAR-10 train set (50K examples) and $D_d$ is the CIFAR-10 test set (10K examples) in our experiments.

### H.2 Attacker objectives.

We identify 3 attacker objectives representing varying levels of privacy violation, ordered by strictly decreasing attack strength.

1. *(Total reconstruction).* The attacker is able to accurately reconstruct every example in $E(D_d)$.
2. *(Partial reconstruction).* The attacker is able to identify a small subset of $E(D_d)$, on which it can accurately reconstruct.
3. *(Set-level reconstruction).* The attacker is able to produce a set of reconstructions and a small subset of them correspond to real training points.

Although a weaker attack than (1), (2) still constitutes a strong privacy violation, since privacy is a *worst-case* notion: a priori, a user submitting their data does not know whether they are part of the reconstructable set or not (see (Carlini et al., 2022) for discussion). Success in (3) would imply an attacker can generate accurate reconstructions, but are unable to identify which candidates correspond to real training data. Note that success in (3) combined with a good membership inference attack implies success in (2).

### H.3 Experimental details

We train U-Net-based conditional denoising diffusion models on the CIFAR-10 train set. During training, extracted features are added to time-step embeddings and condition the model via shift-and-scale operations inside normalization layers. We sample with DDIM and classifier-free guidance.

We find that there is a non-trivial overlap between the CIFAR-10 test and train sets; on which our models memorize and reconstruct perfectly. To account for this, we filter the CIFAR-10 test set for near-duplicates: we remove the 1K test images with the highest SSIM score with a member of the train set, leaving 9K images to evaluate on. We also manually inspect reconstructions and verify that they differ from the closest training image to the target.

### H.4 Results

**Raw feature reconstruction.** We present the results of our reconstruction attack on raw features in Figure 12 for 3 selection methods (*all*, *attacker*, and *oracle*). Each selection method is representative of performance in the corresponding attacker objectives of *total*, *partial*, and *set-level* reconstruction. Quantitative image similarity measures are reported in Table 8.

For *attacker-selection*, we sample 10 reconstructions from each embedding, compute the average pairwise SSIM amongst our reconstruction, and select the top 1% of reconstructions according to this metric. This is

based on the intuition that stronger determinism during sampling implies the model is more confident about what the information in the embedding corresponds to.

| Method | Selection | Similarity measure | | |
|---|---|---|---|---|
| | | *PSNR* ↑ | *LPIPS* ↓ | *SSIM* ↑ |
| ResNet-50 reconstruction | All | 13.5 | .384 | .168 |
| | Attacker | 14.7 | .297 | .358 |
| | Oracle | 16.5 | .257 | .535 |
| MAE reconstruction | All | 15.6 | .305 | .276 |
| | Attacker | 17.9 | .191 | .579 |
| | Oracle | 19.5 | .181 | .674 |
| Train set | Oracle | 14.7 | .389 | .500 |

**Table 8:** Similarity metrics between original image and **raw feature reconstructions** on the filtered CIFAR-10 test set. Figures are computed on different selections of the test set, representing different attacker objectives. **All:** averaged result over entire test set; performance corresponds to *total reconstruction* objective. **Attacker:** attacker selects 1% ($n = 90$) reconstructions without access to ground truth; corresponds to *partial reconstruction* objective. **Oracle:** top 1% ($n = 90$) reconstructions selected based on ground-truth SSIM; corresponds to *set-level reconstruction*.

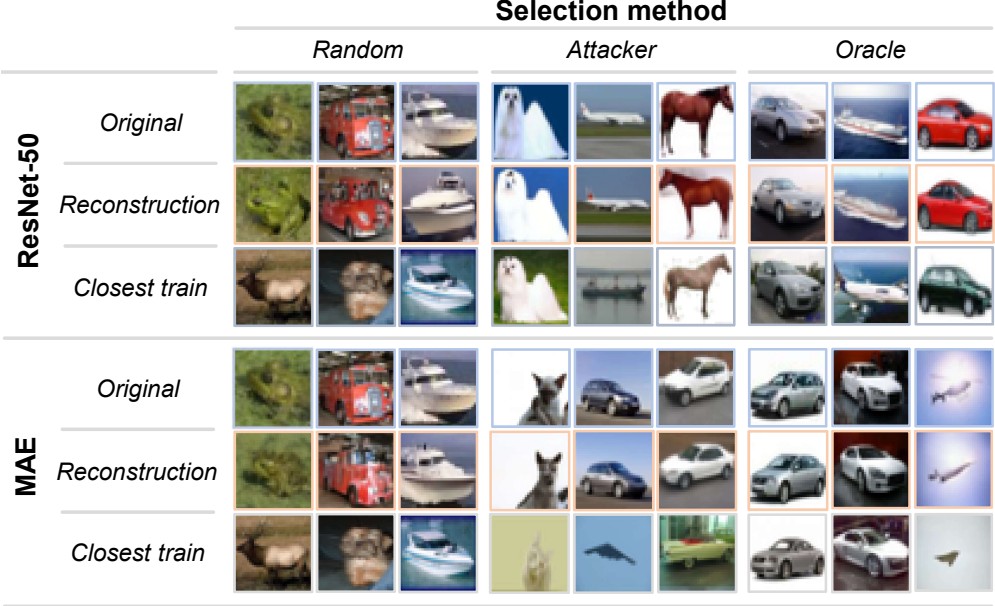

**Figure 12:** Results comparing the original image and **raw feature reconstructions**, along with the original images' closest train set member by SSIM. We present results for 2 backbones: ResNet-50 and Masked Autoencoder; and 3 selection methods (*random*, *attacker*, and *oracle*), which correspond to performance on *total*, *partial*, and *set-level* reconstruction.

Our main result is that an attacker is capable of producing reasonable reconstructions in all 3 settings, outperforming a baseline of selecting the closest training image. Furthermore, we find that: (1) The feature backbone affects reconstruction quality (MAE reconstructions are better than ResNet-50); (2) The attacker can effectively employs heuristics (intra-sample similarity) to identify which reconstructions are likely to be good, approaching the results of oracle selection based on the ground-truth image.

**FedPFT reconstruction.** Table 9 and Figure 13 show results for set-level reconstruction of GMM-sampled features, for both random and worst-case test images. We see that even when with a ground truth similarity oracle, most images fail to be reconstructed (*Oracle-Random* column of Figure 13). DP further diminishes attack effectiveness, in particular for worst-case set-level reconstructions.

| Backbone | $\epsilon$ | Selection | Similarity measure | | |
|---|---|---|---|---|---|
| | | | PSNR ↑ | LPIPS ↓ | SSIM ↑ |
| ResNet-50 | $\infty$ | Oracle | 14.7 | .423 | .508 |
| | | Oracle-all | 12.7 | .546 | .326 |
| | 10 | Oracle | 14.6 | .511 | .483 |
| | | Oracle-all | 12.7 | .571 | .322 |
| MAE | $\infty$ | Oracle | 14.5 | .456 | .497 |
| | | Oracle-all | 12.5 | .558 | .323 |
| | 10 | Oracle | 13.4 | .553 | .452 |
| | | Oracle-all | 12.2 | .595 | .306 |

**Table 9:** Similarity metrics between original images and **set-level reconstructions** on the filtered CIFAR-10 test set. For each test image, we match it to its closest in terms of SSIM among all reconstructions. **Oracle:** the top 1% ($n = 90$) of matched reconstructions by ground-truth SSIM, corresponding to performance on *set-level reconstruction*. **Oracle-all:** average similarity between test images and their matched pairs.

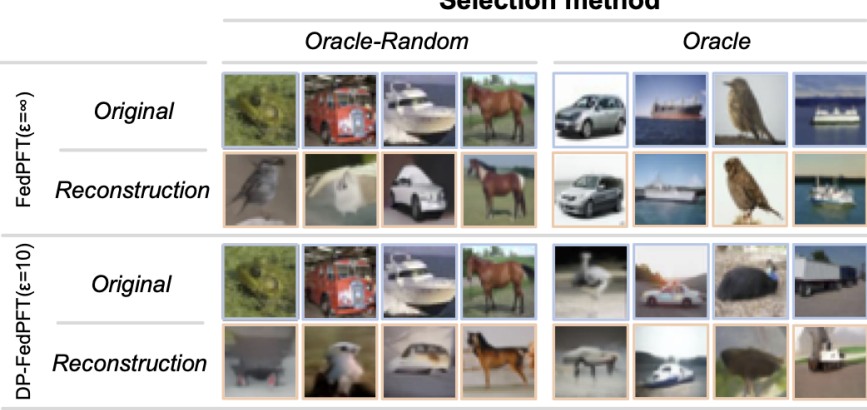

**Figure 13:** Results comparing the original image and **set-level reconstructions** with ResNet-50 backbone. We present results for two selection methods. **Oracle-Random:** the closest image in the reconstruction set by SSIM for random test images. **Oracle:** the closest image in the reconstruction set by SSIM for worst-case test images.

