# OpenReview forum: "Foundation Models Meet Federated Learning: A One-shot Feature-sharing Method with Privacy and Performance Guarantees"
_TMLR — Accepted by TMLR_

### Review · Reviewer_tydh · 2025-02-20

**Summary Of Contributions:**

This paper aims to adapt large foundation models for downstream tasks in Federated Learning (FL) settings while maintain privacy preservation. To solve the challenges of FL’s iterative training and model transmission, this paper introduces FedPFT, a one-shot FL method with a server-side performance bound to reduce communication costs and GPU memory requirements.

In FedPFT, without requiring training or fine-tuning of large foundation models, each client learns and transfers parametric models from extracted features from pre-trained, frozen foundation models in one round. These parametric models are used to generate synthetic features at the server to train a classifier head. This work performs some experiments on accuracy, communication costs, and privacy validation for differential privacy against reconstruction attacks.

**Audience:**

Yes

**Broader Impact Concerns:**

The authors discussed the social impact of distributing foundation models to FL and using differential privacy. It would be better to consider the up-to-date foundation models as baselines for broader impact in the real world.

**Claims And Evidence:**

Yes

**Requested Changes:**

1. Comparing FedPFT with recent works would align your evaluation with current advancements. Please update your experiments accordingly.

2. The experiments (e.g., Figure 4 with 50 clients) lack parameter diversity. Test additional client counts (e.g., 5, 10, 20, 100), explore Gaussian mixture effects on accuracy, and vary DP parameters ($\epsilon, \delta$, noise scale) with comparisons to other DP protections. This will demonstrate scalability and robustness. Also, clarify the vague Theorem 4.1 with precise details or empirical support.

3. Section 4.2’s DP discussion lacks a novel conclusion. Please specify if your framework improves the DP-bound over existing methods, providing a concrete theoretical or empirical outcome to enhance its contribution.

4. Using ResNet-50 (Section 5.1) feels outdated. Incorporating modern models like large language models (LLMs) would better reflect state-of-the-art trends. It would be better to update your experiments with the recent models.

5. It would be better to enrich the justification for shifting to distributed settings with specific motivations.

6. Minor: the font sizes of Figure 6 and Figure 8 are different.

**Strengths And Weaknesses:**

### Strengths
+ The topic of distributing the foundation model is timely and interesting. As large-scale foundation models become increasingly central to various applications, exploring their downstream tasks for their efficient usage seems to be a reasonable need in the field. This focus aligns well with current trends toward scalable and adaptable model deployment.
+ The usage of differential privacy (DP) can improve the privacy guarantee. By incorporating DP, the work safeguards sensitive client data, and the inclusion of experimental results demonstrating the reconstruction attacks further validate the feasibility and necessity of this privacy-preserving mechanism.
+ The idea can reduce communication costs by avoiding transmitting the original client-side model. This optimization reduces communication and computational overhead, making the framework more viable for distributed systems where resource efficiency is critical.

### Weaknesses

- The experiments benchmark the proposed method against somewhat dated works, such as FedKT and FedBE, both published in 2020. To strengthen the evaluation, comparisons with more recent studies—such as "FedFed: Feature Distillation against Data Heterogeneity in Federated Learning"—would provide a more current and relevant context, especially given the rapid evolution of federated learning and feature sanitization techniques.

- The experimental setup lacks diversity in parameter choices. For instance, Figure 4 considers only 50 clients for comparison. Including experiments with varying client counts (e.g., 5, 10, 20, or 100) could better demonstrate the framework’s scalability. Additionally, the impact of Gaussian mixture parameters on accuracy remains unexplored, as does the effect of varying DP parameters (e.g., different $\epsilon$ or noise scale) Comparing these with other DP-based protections in federated learning could further clarify the method’s effectiveness. Moreover, Theorem 4.1 appears vague and would benefit from more precise articulation or empirical support.
- The differential privacy analysis seems not to derive a new concrete conclusion of the proposed framework. In Section 4.2, the authors utilize DP for preserving privacy. However, it seems that the author did not give a new concrete conclusion of DP guarantee? It is unclear whether the proposed framework improves the DP-bound compared to existing methods. A clearer explanation of how (or if) this work advances DP guarantees would strengthen its theoretical contribution. Besides, the analysis to Gaussian mechanism seems to be not new. A better clarification is required to demonstrate its theoretical contribution.
- The experiments rely on older foundation models like ResNet-50 (Section 5.1), which may not reflect the state-of-the-art. Incorporating more contemporary models, such as large language models (LLMs), could better align the work with current research trends and demonstrate its applicability to cutting-edge systems.
- The rationale for distributing foundation models in a decentralized framework is not sufficiently justified. LLM is very powerful in a centralized setting. The more explicit motivation to distribute it should be explained more, particular in specific advantages or necessities of adapting them to distributed environments—such as privacy, scalability, or resource constraints.

---

> ### Author Response · Authors · 2025-03-19
> **Response to Reviewer tydh**
>
> We thank the reviewer for their thorough comments. We are pleased that they find the foundation model topic interesting and our use of differential privacy (DP) and analysis of communication cost well-motivated.
>
> 1. **New Baselines**: We appreciate the reviewer’s suggested baseline. Our primary focus has been on comparing our method with one-shot FL approaches, such as FedNCM (Legate et al., 2024). In total, we have evaluated our work against eight baselines. We will add a comparison with FedFed in the final manuscript.
>
> 2. **Effect of Parameters**: Number of clients: We have evaluated FedPFT across different client settings—50 clients in Figure 4, 3 clients in Table 3, and 5 clients in Figure 5—demonstrating its effectiveness. Across all settings, FedPFT consistently achieves within 5% of centralized performance, indicating its robustness to varying client numbers.
> Privacy budget (ε): We are currently conducting an experiment to analyze the effect of ε and will include the results in the final manuscript.
>
> 3. **Foundational Models**: As shown in Table 2, we use Vision Transformers trained on ImageNet, CLIP Vision Transformers, and ResNet-50 as foundational models. As discussed in the limitations section (A.6), we focus on vision tasks in this work and leave the extension to NLP tasks, including next-word prediction, for future work.
>
> 4. **Motivation for the Decentralized Setting**: We appreciate the reviewer’s comments and have updated the manuscript to strengthen our motivation:
> Traditional FL relies on a central server for coordination, which introduces a single point of failure and limits scalability. This centralized architecture also poses privacy and security risks by exposing model updates and reduces fault tolerance by making the system dependent on server availability. Decentralized FL mitigates these issues by enabling direct peer-to-peer communication, improving resilience, and eliminating the need for a trusted aggregator. Therefore, we extend FedPFT to the decentralized setting.
>
> 5. **Differential Privacy Analysis**:The purpose of the DP section is to demonstrate how DP can be applied to FedPFT, providing formal privacy guarantees. Theorem 4.1 establishes that by adding the specified noise, FedPFT satisfies (ϵ,δ)-differential privacy.
>
> Please let us know if you have any further comments.

---

### Review · Reviewer_ncqR · 2025-03-01

**Summary Of Contributions:**

This paper proposes a novel one-shot (i.e. one communication round) Federated Learning (FL) method FedPFT leveraging foundation models. In its centralized version, a task-specific frozen foundation model is employed to extract feature representations from client data. These features are subsequently modeled using Gaussian Mixture Models (GMMs) to capture class-conditional distributions. The clients then transmit the GMM parameters along with corresponding class information to the server, where the distributions are merged and utilized to train a classification head. Empirical studies have shown that centralized FedPFT achieves more accurate prediction than other one-shot FL methods while maintaining comparable communication costs. Moreover, private and decentralized learning can also be considered in this framework. The theory supports the proposed method by giving performance and privacy guarantees.

**Audience:**

Yes

**Broader Impact Concerns:**

I have no ethical concerns.

**Claims And Evidence:**

Yes

**Requested Changes:**

1. Clarification on the difference between FedPFT and CCVR.
2. Include motivations on the choice of $K$ for DP-FedPFT.
3. Provide more details in Figure 8.

**Strengths And Weaknesses:**

**Strengths:**

1. The paper is mostly well-written and easy to read.
2. The method is well-motivated. The authors provided sufficient evidence to show that GMMs effectively approximate the distribution of the extracted features from foundation models (and the advantage of increasing $K$ in performance). Moreover, the true loss of the model can be estimated by the loss trained on the synthetic features.
3. Variants of the method can adapt to decentralized FL and private learning settings, which is not possible for existing one-shot methods.
4. Extensive experiments are provided to support the authors' claims, including high accuracy, comparable communication cost, and good performance against reconstruction attacks.

**Weakness:**

1. I think the comparison between FedPFT and CCVR in Section 2 should be more extensive to highlight the contributions and differences. In particular, CCVR learns the features and the Gaussian approximation simultaneously through multi-round communication and finally retrains the model on the learned Gaussian representations. FedPFT uses a pre-trained model to extract the features and learns the GMM approximation for server training. So I would expect CCVR using a pre-trained model to perform similarly to FedPFT ($K=1$), is that correct? Although I acknowledge that increasing $K$ can largely enhance the model performance.
2. It would be better to motivate the choice of $K$ in the differentially private version (currently $K=1$). Is it due to the theory or the computational cost?
3. In Figure 8, the author showed that FedPFT is robust to data heterogeneity, but it is not clear which $K$ is used and whether the choice of $K$ is sensitive.
4. Some minor errors:
* Equation 1, Using $\frac{n_i}{n}$ as the coefficient is more common in the FL literature.
* Section 4.1, line 5 "Next, each client learns runs the Expectation ..."
* Page 14, "A.2 Does sending sending label counts ... "

---

> ### Author Response · Authors · 2025-03-19
> **# Response to Reviewer ncqR**
>
> We thank the reviewer for their thorough analysis of our work. We are pleased that they find our paper well-written and well-motivated. We also appreciate their recognition of our algorithm's support for decentralized and private settings, which we find particularly exciting as well.
>
> Below, we address the reviewer’s comments:
>
> 1. **Comparison between FedPFT and CCVR:** CCVR, when using a pre-trained model, does not perform similarly to FedPFT (K=1). This is because, in FedPFT, we train the classifier head from *scratch* using Gaussian mixtures, whereas in CCVR, clients send a trained classifier head to the server, which then post-calibrates (*fine-tunes*) the classifier at the end of each round to mitigate bias and improve accuracy in multi-round FL. In Figure 4, we compare CCVR with a frozen pre-trained model against FedPFT, showing that FedPFT outperforms CCVR while also reducing communication costs by avoiding classifier head transmission. We believe that the bias in CCVR’s local classifier heads prevents it from achieving competitive performance compared to FedPFT. Furthermore, as shown in Table 1, CCVR does not provide any analysis of its performance. We have incorporated this discussion into the paper’s discussion section.
>
> 2. **Choice of K in the DP setting**: We selected K=1 for our DP setting since our theoretical analysis currently supports only this case. Extending DP to K>1 is nontrivial, and we leave it for future work. We have updated the manuscript to clarify this reasoning.
>
> 3. **Choice of K in Figure 8**: We have updated the manuscript to specify that we use the lowest values of K (i.e., K=1) and spherical Gaussians. Additionally, we highlight that FedPFT’s results are not sensitive to the choice of K, as demonstrated in Figure 4.
>
> 4. **Minor errors**: Thank you for pointing out these errors. We have updated the manuscript to correct them.
>
> Please let us know if you have any further questions.

---

### Review · Reviewer_bpfy · 2025-03-10

**Summary Of Contributions:**

The authors propose FedPFT, which incorporates a one-shot federated learning approach based on transmitting Gaussian Mixture Model parameters. This method addresses the communication cost and GPU memory limitations typical of conventional FL in foundation models. FedPFT offers significant performance guarantees, including scalability (both decentralized and centralized), privacy protection, and robustness to data heterogeneity. The authors support these claims with both theoretical proofs and empirical results across a variety of experimental setups.

**Audience:**

Yes

**Claims And Evidence:**

Yes

**Requested Changes:**

See the above.

**Strengths And Weaknesses:**

Strengths
- The writing is clear, and the extensive experiments conducted in diverse setups are convincing. Theoretical results further substantiate the research, giving it a concrete foundation.

Weaknesses
- However, the paper primarily lists experimental achievements and lacks deeper analytical insights. For instance, providing intuition on why FedPFT performs better with certain covariance types and why it remains agnostic to data heterogeneity would be beneficial.
- It would also enhance the work to connect the high accuracy of FedPFT more directly with the low-level analysis presented in Theorem 4.2.—for example, by incorporating the design of the intended loss and presenting the corresponding accuracy graph.
- I cannot find the architecture used for the attacker model.
- The following sentence appears to be duplicated:
"Also, Figure 4 shows different tradeoffs for different numbers of mixtures K and covariance types. Additionally, Figure 4 demonstrates different tradeoffs for varying numbers of mixtures K and covariance types."

---

> ### Author Response · Authors · 2025-03-19
> **# Response to Reviewer bpfy**
>
> We thank the reviewer for their helpful comments. We are pleased that they find our experiments extensive and our theoretical results foundational.
>
> Below, we address the reviewer’s comments:
>
> 1. **Additional insights into FedPFT:** We appreciate the reviewer’s suggestion and have updated the manuscript to reflect the following:
> The strong performance of FedPFT stems from its use of pre-trained foundational models to extract task-agnostic features. Additionally, GMMs require only a few parameters (centroids and variances) from these features, allowing them to perform well even with limited client data. Furthermore, since FedPFT recreates features at the server, it remains agnostic to data heterogeneity. Increasing GMM complexity—from spherical to diagonal and full covariance—enables better feature representation at the cost of higher communication overhead.
>
> 2. **Architecture of the attacker model:** We provide details of our attack experiments in Section H of the appendix. Specifically, we use a U-Net-based conditional denoising diffusion model trained on the CIFAR-10 dataset. We have updated the manuscript to include this information in the main text.
>
> 3. **Connecting Theorem 4.2 with the loss**: Theorem 4.2 establishes that the centralized loss is bounded by the EM loss of fitting a GMM. This result suggests that increasing the number of learnable parameters in GMMs (e.g., using diagonal instead of spherical covariance) improves performance. This finding is further validated by our experimental results in Figure 6.
>
> 4. **Typing errors:** Thank you for pointing out these errors. We have corrected them in the manuscript.
>
> Please let us know if you have any further comments.

---

### Public Comment · ~Marco_Ciccone1 · 2025-07-08
**Nice work and stronger baselines**

Congrats on the nice work!!

It would be interesting to try stronger baselines than FedNCM using second-order statistics and extend them with privacy guarantees.

Our Fed3R algorithm [1] incrementally computes a Ridge Regression classifier and could be a strong candidate.
As you can see, the algorithm has much better performance than FedNCM and is robust in realistic case scenarios as Inaturalist and gldv2.

[1] Fanì, E., Camoriano, R., Caputo, B., & Ciccone, M. Accelerating heterogeneous federated learning with closed-form classifiers. ICML, 2024

**Code:** https://github.com/Erosinho13/Fed3R

Best,

Marco Ciccone

---

### Decision · Action_Editor_eVZG · 2025-04-23

**Recommendation:** Accept with minor revision

**Comment:**

This paper proposes a novel one-shot (i.e. one communication round) Federated Learning (FL) method FedPFT leveraging foundation models. It is well-written and motivated, providing a model featuring good performance and Differential Privacy (DP) guarantee. The reviewers suggest the authors to include a few more experiments in the final version of the paper: experiments with different differential privacy budget $\epsilon$, a comparison with the baseline FedFed.

**Audience:**

Yes

**Claims And Evidence:**

Yes